# VisDiff: SDF-Guided Polygon Generation for Visibility Reconstruction, Characterization and Recognition

**Rahul Moorthy**
University of Minnesota
mahes092@umn.edu

**Jun-Jee Chao**
University of Minnesota
chao0107@umn.edu

**Volkan Isler**
The University of Texas at Austin
isler@cs.utexas.edu

## Abstract

The ability to capture rich representations of combinatorial structures has enabled the application of machine learning to tasks such as analysis and generation of floorplans, terrains, images, and animations. Recent work has primarily focused on understanding structures with well-defined features, neighborhoods, or underlying distance metrics, while those lacking such characteristics remain largely unstudied. Examples of these combinatorial structures can be found in polygons, where a small change in the vertex locations causes a significant rearrangement of the combinatorial structure, expressed as a visibility or triangulation graphs. Current representation learning approaches fail to capture structures without well-defined features and distance metrics.

In this paper, we study the open problem of *Visibility Reconstruction*: Given a visibility graph $G$, construct a polygon $P$ whose visibility graph is $G$. We introduce **VisDiff**, a novel diffusion-based approach to generate polygon $P$ from the input visibility graph $G$. The main novelty of our approach is that, rather than generating the polygon's vertex set directly, we first estimate the signed distance function (SDF) associated with the polygon. The SDF is then used to extract the vertex location representing the final polygon. We show that going through the SDF allows **VisDiff** to learn the visibility relationship much more effectively than generating vertex locations directly. In order to train **VisDiff**, we create a carefully curated dataset. We use this dataset to benchmark our method and achieve **26%** improvement in F1- Score over standard methods as well as state of the art approaches. We also provide preliminary results on the harder visibility graph recognition problem in which the input $G$ is not guaranteed to be a visibility graph. To demonstrate the applicability of VisDiff beyond visibility graphs, we extend it to the related combinatorial structure of triangulation graph. Lastly, leveraging these capabilities, we show that VisDiff can perform high-diversity sampling over the space of all polygons. In particular, we highlight its ability to perform both polygon-to-polygon interpolation and graph-to-graph interpolation, enabling diverse sampling across the polygon space.

## 1 Introduction

Polygons are widely used as geometric representations in domains such as cartography [1], architectural design [2], and robotic motion planning [3]. Applications across these domains require understanding the combinatorial structure of polygons for analysis, reasoning, and generation. The combinatorial structure captures the discrete relationships between polygon vertices, independent of their specific coordinates, angles, or edge lengths. A key example of such a structure is the visibility graph (Appendix A), which encodes mutual visibility between regions. Visibility graphs enable privacy-aware floorplan design [2] and the extraction of topographic features in terrain analysis [4]. While generative models are increasingly used in these applications [5]. Most existing

39th Conference on Neural Information Processing Systems (NeurIPS 2025).

approaches [6, 7] rely solely on geometric coordinates. Despite the structural importance of visibility graphs, they have not been leveraged to guide the generative process. To address this gap, we investigate representations that link polygons to their combinatorial structures.

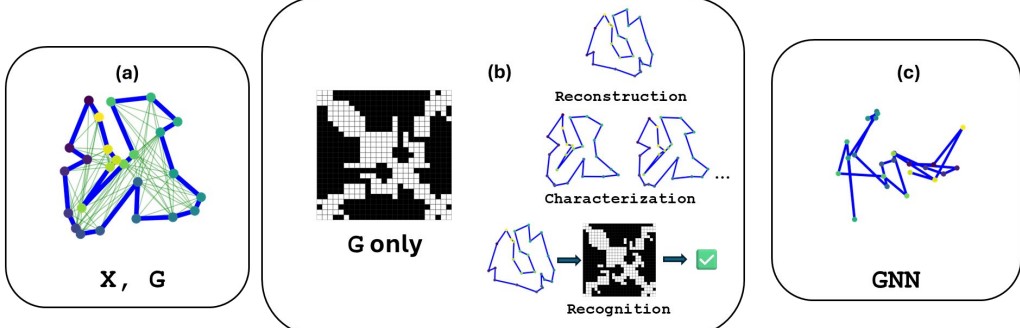

Figure 1: **a)**: A polygon P is given by an ordered list of vertex locations $X$. Also shown are the visible edges of the polygon in green. **b)**: The visibility graph $G$ of polygon P represented as an adjacency matrix where black denote non-visible edge while white denote visible edges. We seek to answer the question: How much information about $X$ can be recovered from $G$ alone? We show the output of VisDiff for the reconstruction, characterization and recognition problems associated with $G$. **c)**: GNN output of $G$ for reconstruction problem. Clearly, standard GNN embedding methods are not sufficient to recover the vertex locations $X$ from $G$.

The main questions we study are as follows: Suppose we are given the visibility graph $G$ of a polygon, as shown in Figure 1b. Note that $G$ contains no coordinate information from $X$. What can we say about the polygon (Reconstruction) and the set of all polygons (Characterization) that have this graph $G$ as their visibility graph. Can we determine whether a polygon exists for $G$ (Recognition). Given the recent success of learning-based graph-embedding approaches, it might be tempting to apply GNN-based methods [8] directly to reconstruct $X$. However, since $G$ lacks a natural distance metric that such models can exploit, these methods fail to directly predict vertex coordinates from $G$ (Figure 1c).

In this paper, to overcome the lack of distance information in $G$ for polygon generation, we present **VisDiff**: a generative diffusion model that employs an intermediate signed distance function (SDF) representation. Specifically, given a visibility graph $G$ as input, VisDiff generates the SDF of the corresponding polygon. The vertex locations are then extracted from the zero level set of the SDF, yielding the polygon associated with $G$. Our core contribution is that using the SDF as an intermediate representation enables the generation of meaningful polygons that preserve visibility constraints and provide strong evidence for all characterization, reconstruction, and recognition as shown in Figure 1b. This contrasts with prior approaches that rely directly on vertex [8] or triangulation [9] representations. To train VisDiff, we construct a carefully curated dataset that captures a broad range of polygon combinatorial properties. Existing random polygon generation methods struggle to faithfully represent the visibility graph space, often biasing toward high concavity as the number of vertices increases. We address this issue by systematically rebalancing the dataset by link diameter, which quantifies concavity. The dataset is made publicly available for further research.

We demonstrate the generality of VisDiff beyond visibility graphs. Specifically, we apply it to the task of reconstructing a polygon from its triangulation graph. Finally, we leverage these capabilities to highlight its ability to perform high-diversity sampling over the space of all polygons, making it suitable for data augmentation in applications involving polygon representations.

In summary our key contributions are:

- We initiate a learning based study of polygon reconstruction and characterization problems based on combinatorial structures without well-defined distance and neighborhood. We show that existing state-of-the-art approaches fail to effectively capture the connection between combinatorial and geometric properties.
- We design a carefully curated dataset that captures a wide range of combinatorial properties of polygons and make it publicly available for further research.

- We present **VisDiff**: a generative diffusion model that generates an intermediate SDF representation corresponding to G, which is then used to extract the polygon. We show that using SDF as an intermediate representation yields meaningful polygons for G. We evaluate VisDiff with baselines on *Visibility Reconstruction* and demonstrate its capability to perform *Visibility Characterization*. Additionally, we also provide preliminary results for *Visibility Recognition*.
- We demonstrate the generality of **VisDiff** to both visibility graph and triangulation graph based polygon generation.
- We leverage all these capabilities to highlight the ability to perform high diversity sampling over the space of all polygons making it suitable for applications such as data augmentation.

## 2 Related Work

We summarize the related work in three directions: theoretical results for visibility graph reconstruction and recognition, representation learning for shapes, and graph neural networks for polygons.

**Visibility Graph Reconstruction and Recognition:** The problems of reconstructing and recognizing visibility graphs have been studied extensively in the theoretical computational geometry literature, yet they remain open [10]. Existing results address reconstruction and recognition only for specific polygon categories, including pseudo [11], convex fan [12], terrain [12], spiral [13], anchor [14], and tower [15] polygons. On the hardness side, the visibility graph recognition and reconstruction problems are known to lie in PSPACE [16], specifically within the Existential Theory of the Reals class [17]. However, the exact computational hardness of these problems remains unresolved. In this work, we explore them from a representation learning perspective, investigating whether generative models can learn the underlying manifold of the space of polygons and their visibility graphs in a generalizable manner.

**Representation Learning:** 3D shape completion [18, 19, 20, 21] is a closely related application. In 3D shape completion, the input typically contains partial geometric information, such as a point cloud. In contrast, our input consists solely of a combinatorial description, namely the visibility graph. Multiple shapes may be consistent with the same input graph, and recovering them without any geometric cues is a fundamentally challenging task. Another related body of work is mesh generation [6, 7]. Recent advances in this area include MeshGPT [22], MeshAnything [9], and PolyDiff [23]. These approaches generate high-quality 3D triangular meshes by learning to output a set of triangles from a fixed set of triangles. PolyDiff discretizes the 3D space into bins, whereas MeshAnything and MeshGPT operate over a predefined set of triangles. In contrast, our work seeks to learn the continuous space of all polygons and their corresponding visibility graphs.

**Graph Neural Networks (GNNs):** GNNs are a standard class of models used for learning on graph-structured data. Most existing work focuses on graphs with features embedded in a well-defined metric space. The closest to our setting is the generation of graph embeddings from distance matrices. Cui et al. [24] proposed MetricGNN, which learns graph embeddings from a given embedding distance matrix. Yu et al. [25] introduced PolygonGNN, which effectively represents multi-polygon data for graph classification tasks by leveraging visibility relationships between polygons. Specifically, PolygonGNN demonstrated that augmenting vertex embeddings of individual polygons with both spatial location information and visibility relationships to other vertices leads to improved geometric representation learning. All of the above approaches assume the existence of an underlying metric space or spatial position information, both of which are absent in the visibility graph reconstruction problem. We develop **VisDiff** to learn meaningful embeddings in this challenging combinatorial domain.

## 3 Problem Formulation

We only study polygons which are simple (the boundary does not self intersect) and simply-connected (no holes). Let $X \in \mathbb{R}^{N \times 2}$ be the $N$ vertex locations of a polygon $P$, $G \in \mathbb{R}^{N \times N}$ be the adjacency matrix representing visibility graph of $P$, $Vis(P)$ be a function to determine the visibility graph of P as an adjacency matrix. We consider the following problems of increasing difficulty:

**Problem 1 (Reconstruction)** *Given a valid G, generate **a** polygon P such that $Vis(P) = G$.*

**Problem 2 (Characterization)** *Given a valid $G$, generate **all** polygons $P$ such that $Vis(P) = G$.*

Note that in these two problems, the input $G$ is assumed to be valid – i.e., there exists a polygon $P$ whose visibility graph is $G$. We also formulate a more general recognition problem in which $G$ is arbitrary:

**Problem 3 (Recognition)** *Given an arbitrary graph $G$, determine whether there exists a polygon $P$ such that $Vis(P) = G$.*

## 4 Method

We present VisDiff for generating a polygon distribution given the visibility graph. VisDiff models the polygon distribution using diffusion models [26] conditioned on the input graph. The key idea is to use the signed distance function (SDF) as an intermediate representation for polygon generation. In this section, we first provide a brief background on diffusion models, then introduce the two main components of VisDiff: (i) a graph-conditioned diffusion model for SDF generation, and (ii) a polygon vertex extraction module that reconstructs the polygon from the predicted SDF. Detailed architecture specifications are provided in Appendix J.

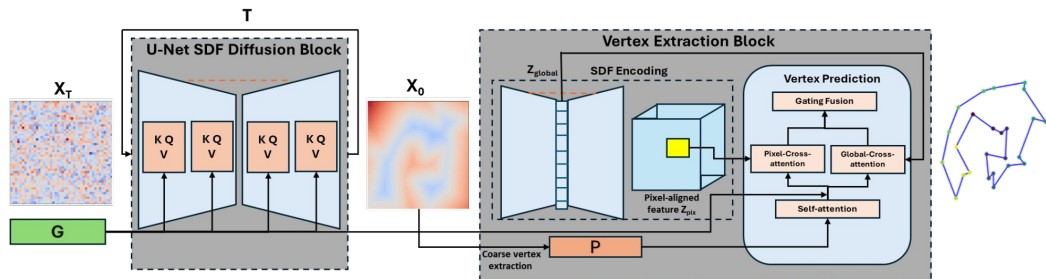

Figure 2: VisDiff architecture. The model consists of two main components: the U-Net SDF Diffusion block and the Vertex Extraction block. **U-Net Diffusion Block:** A noisy SDF, denoted as $\mathbf{X}_T$, is first sampled from a Gaussian distribution. $\mathbf{X}_T$ then passes through $\mathbf{T}$ timesteps of the reverse diffusion process to generate the clean SDF $\mathbf{X}_0$. This denoising process is conditioned on the input graph $\mathbf{G}$ using transformer cross-attention blocks represented by $\mathbf{K}$, $\mathbf{Q}$, and $\mathbf{V}$, which correspond to the key, query, and value terms, respectively. In our approach, $\mathbf{Q}$ is obtained from the learned spatial CNN features, while $\mathbf{K}$ and $\mathbf{V}$ are derived from $\mathbf{G}$. An initial set of vertices $\mathbf{P}$ is then estimated from $\mathbf{X}_0$ via contour extraction. **Vertex Extraction Block:** Given the predicted SDF $\mathbf{X}_0$, the SDF encoder generates pixel-aligned features $Z_{pix}$ and global features $Z_{global}$. These features, along with the initial vertices $\mathbf{P}$, are fed into the vertex prediction block to predict the final vertex locations. During **Training**, the model is supervised using both the ground-truth SDF and the corresponding polygon. During **Testing**, only the visibility graph $\mathbf{G}$ is provided as input.

### 4.1 Background

Diffusion models have demonstrated the ability to efficiently map a Gaussian distribution to a target data distribution [26, 27, 28, 29]. The Denoising Diffusion Implicit Model (DDIM) [30] consists of two main stages: the forward diffusion process and the reverse denoising process.

*Forward diffusion* involves progressively adding noise to the data according to a predefined schedule. Let the data sample from the target distribution be denoted as $x_0$. At diffusion timestep $t$, given a noise standard deviation $\sigma_t > 0$, the noisy sample is defined as

$$x_t = x_0 + \sigma_t \epsilon, \tag{1}$$

where $\epsilon \sim \mathcal{N}(0, I)$ is Gaussian noise. In this way, noise is gradually injected into the data until it becomes a pure Gaussian sample at the end of the forward process. VisDiff employs a linear-log scheduler [31] to control the noise level throughout this process.

*Reverse diffusion* involves recovering the original data from the final Gaussian sample produced during the forward process. In this step, we start with Gaussian noise and iteratively predict the noise

added to the sample given $\sigma_t$. The reverse process is parameterized by a neural network that learns to predict the added noise from the noisy sample and the corresponding $\sigma_t$.

## 4.2 Graph conditioned SDF diffusion

The core idea of VisDiff is to predict the polygon distribution conditioned on the input graph. However, such a graph contains no coordinate information, posing a challenge for existing methods that attempt to directly predict polygon vertex coordinates. To obtain the ground-truth SDF, we first normalize each polygon to fit within a unit square and compute the SDF values on a $40 \times 40$ grid, resulting in an image where each pixel stores the distance to the nearest point on the polygon boundary. In this manner, polygons are represented by their signed distance functions as images.

We adopt an architecture similar to Latent Diffusion [32] due to its ability to produce high-quality generations under external conditioning. However, unlike Latent Diffusion, we train directly on the SDF rather than on latent features. Specifically, we use a time-conditioned U-Net [33] encoder–decoder architecture to predict the noise added to the original SDF sample. The U-Net CNN blocks are conditioned on the encoded visibility graph using Spatial Transformer Cross-Attention [34] blocks, which integrate visibility information into the U-Net's spatial features during training. The key and value components of each cross-attention block correspond to the lower-triangular part of the visibility adjacency matrix, since it is symmetric in nature, while the queries are derived from the spatial CNN features. Figure 2 illustrates the architecture of the SDF diffusion block. The model is trained using $L_{MSE}$ mean-squared error loss between the predicted and true noise added to the sample. Given a visibility graph $G$, the trained model is then used to sample polygon SDFs.

*Sampling* of the SDF is performed using a DDIM sampler. The process begins by drawing a sample $x_t \in \mathbb{R}^{40 \times 40}$ from a Gaussian distribution $\mathcal{N}(0, I)$, followed by a schedule of decreasing noise levels proportional to the number of diffusion steps. Each reverse step is defined as

$$x_{t-1} = x_t + (\sigma_{t-1} - \sigma_t)\, \epsilon_\theta(x_t, \sigma_t, G), \tag{2}$$

where $\epsilon_\theta(x_t, \sigma_t, G)$ denotes the noise predicted by the U-Net encoder–decoder architecture, conditioned on the visibility graph $G$, the current noisy sample $x_t$, and the corresponding noise level $\sigma_t$. This iterative process reconstructs the polygon's SDF while preserving the visibility constraints imposed by $G$.

## 4.3 Vertex Extraction

The generated SDF of the polygon is then used to determine the final vertex locations whose visibility relationships correspond to the visibility graph $G$. The process of selecting vertex locations along the zero level set is challenging, as polygon corners are often ill-defined in the SDF representation. Furthermore, as the number of vertex locations increases, a small change in the placement of points on the SDF will significantly alter the visibility structure of the entire polygon.

We formulate polygon vertex extraction as a separate estimation problem: determining vertex locations given the SDF and the visibility graph $G$. The vertex extraction architecture comprises two modules: **SDF Encoding** and the **Vertex Prediction Block**. Figure 2 illustrates the architecture of the vertex prediction block.

**SDF Encoding:** Understanding the fine-resolution structure of the SDF is essential for extracting the underlying polygon representation given $G$, as small perturbations in point placement on the SDF can significantly alter the visibility structure of the entire polygon. Previous continuous-coordinate polygon extraction methods [35, 36, 37] from images typically rely only on global feature extractors. In contrast, pixel-aligned features have proven highly effective for capturing fine-grained details in tasks such as object detection [38] and 3D reconstruction [39]. Since our polygon extraction from SDF requires fine-grained spatial information, we adopt a PIFu-inspired [39] architecture to extract both pixel-level and global features for encoding the SDF. Specifically, we train a U-Net to encode the SDF into a pixel-aligned embedding space $Z_{\text{pix}} \in \mathbb{R}^{40 \times 40 \times 128}$ and a global embedding $Z_{\text{global}} \in \mathbb{R}^{25 \times 512}$. The generated SDF features are then passed to the vertex prediction block to estimate the ordered vertex locations of the polygon.

**Vertex Prediction:** We adopt an architecture similar to Polyformer [40] to generate the final polygon from the encoded SDF. Previous studies have shown that polygon initialization improves localization

accuracy compared to random initialization or the use of learnable queries [37]. Therefore, the vertex locations of the polygon $P$ are first extracted using contour detection methods and simplified to 25 vertices using the area-based simplification technique of Visvalingam [41]. The simplified vertex locations are converted into positional embeddings of size 256, and ordering-based positional embeddings are added to capture cyclic ordering. These vertex embeddings, together with $Z_{\text{global}}$, $Z_{\text{pix}}$, and $G$, are then used to predict the final vertex locations.

Specifically, the vertex embeddings are used as queries $Q$ in three layers of transformer encoders to refine features based on $Z_{\text{global}}$, $Z_{\text{pix}}$, and $G$. Each encoder layer performs self-attention, where the vertex embeddings serve as keys $K$ and values $V$. The self-attention is followed by cross-attention, where $K$ and $V$ are formed by concatenating $Z_{\text{global}}$ and $Z_{\text{pix}}$ with $G$. The pixel-aligned feature map $Z_{\text{pix}}$ is extracted only at the vertex locations $P$ using bilinear interpolation. To adaptively balance the contributions of $Z_{\text{global}}$ and $Z_{\text{pix}}$, we employ a sigmoid-gating fusion mechanism [42] to combine their cross-attention outputs. Figure 2 illustrates the structure of each transformer encoder block. An ablation study (Appendix C) shows that combining pixel-level and global features yields significantly more accurate polygon reconstructions than using global features alone for the visibility reconstruction task.

**Training:** The vertex prediction and SDF encoding blocks are jointly trained using an $L_{\text{MSE}}$ loss, which penalizes deviations of the predicted vertex locations from the ground-truth positions. The vertex extraction and SDF diffusion blocks are trained in two stages. In the first stage, only the SDF diffusion block is trained. In the second stage, the SDF diffusion block is frozen, and only the vertex extraction block is trained. This two-stage training strategy is necessary because contour initialization and simplification are non-differentiable operations, which prevent gradient propagation through the entire architecture during joint training.

## 5   Dataset Generation

The *Visibility Characterization* and *Visibility Reconstruction* problems require a dataset distribution with a key property: multiple polygons $P$ corresponding to the same visibility graph $G$. In addition, the dataset should exhibit high diversity across different visibility graphs. Since no existing dataset satisfies these criteria, we construct one by uniformly sampling polygons based on the graph properties described below and generating multiple augmentations of each polygon.

The dataset generation process involves sampling 60,000 polygons, each with 25 vertex locations arranged in a fixed anticlockwise order. The vertex coordinates are drawn from a uniform distribution within $[-1, 1]^2$. We employ the 2-opt move algorithm [43] to generate valid polygons from the sampled locations. However, the resulting dataset exhibits non-uniformity with respect to the link diameter of the visibility graph, where link diameter quantifies the maximum number of edges on the shortest path between any two graph nodes. A higher diameter indicates greater polygon concavity. To achieve a balanced distribution, we resample the dataset based on the link diameter of the visibility graph, resulting in a final subset of 18,500 polygons. Additional statistics in Appendix B.1 (Figure 5b) show that our dataset is uniformly distributed in terms of link diameter.

We further augment each polygon to generate 20 samples per visibility graph $G$. Shear transformations and vertex perturbations are applied while preserving the visibility graph structure. These augmentations introduce the property of multiple polygons sharing the same combinatorial graph $G$. Both augmentation and resampling are essential for learning the representative space of the *Visibility Characterization* and *Visibility Reconstruction* problems. The final training dataset consists of 370,000 polygons along with their corresponding visibility and triangulation graphs.

The test dataset for validating our approach is generated in two splits: *in-distribution* and *out-of-distribution*. In-distribution samples are obtained by setting aside 100 unique polygons per link diameter from the larger dataset, ensuring they are not included in the training set. Out-of-distribution samples are generated using specific polygon types whose visibility graph properties differ significantly from those in the training set. Appendix B.2 provides details of the out-of-distribution test set generation process and further demonstrates that our dataset exhibits greater diversity in visibility graph properties compared to existing real-world datasets. Both the training and testing datasets are publicly available for further research.

# 6 Experiments

We compare VisDiff with existing methods on the *Visibility Reconstruction* problem and further demonstrate its effectiveness on the *Visibility Characterization* problem. We also present preliminary results for the *Visibility Recognition* problem. In addition, we evaluate the generalization capability of VisDiff to other combinatorial graph structures, such as triangulation graphs. Finally, we demonstrate the ability of VisDiff to perform high-diversity data sampling over the space of all polygons.

## 6.1 Experimental Setup

**Metrics:** To evaluate our algorithm, we compute the visibility graphs of the generated polygons and compare them with the ground truth. We formulate this as a binary classification problem, where each edge in the visibility graph is classified as either *visible* or *non-visible*. We report accuracy, precision, recall, and F1-score between the generated and ground-truth visibility graphs. Each visibility graph is evaluated individually, and the average performance over the dataset is reported as the collective quantitative metric. Since the ratio of visible to non-visible edges can vary significantly across polygons, we primarily use the F1-score to assess model performance.

**Baselines:** We compare VisDiff against baselines capable of conditional polygon generation using either vertex-based or triangulation-based representations. Specifically, we evaluate state-of-the-art approaches including MeshAnything (Mesh) [9], Vertex-Diffusion (VD) [44], Conditional-VAE (C-VAE) [45], and GNN-based generation [46]. We use the publicly available implementations of MeshAnything and Conditional-VAE. MeshAnything is modified to operate on 2D polygon triangulation representations instead of 3D meshes. For the GNN and Vertex-Diffusion baselines, we adopt the architectures from MGNN [46] and PolyDiff [44], respectively. All baselines are trained on our dataset to ensure fairness in evaluation.

## 6.2 Visibility Reconstruction

We demonstrate the ability of VisDiff to learn meaningful polygon representations from visibility graphs on the *Visibility Reconstruction* problem. Table 1 reports the quantitative evaluation on the in-distribution dataset. MeshAnything [9] often generates disconnected triangle sets. Therefore, we report results only for polygons forming a closed polygonal chain. It can be observed that VisDiff outperforms Conditional-VAE and Vertex-Diffusion, which rely on vertex representations, and MeshAnything, which uses a triangulation-based representation, by leveraging the intermediate SDF representation across all metrics. Appendix D (Figure 8) presents additional qualitative comparisons of the baselines with VisDiff. VisDiff learns to generate polygons that closely match the ground-truth visibility, whereas the baselines often produce invalid polygons. Out-of-distribution results, visibility reconstruction selection strategy, and further qualitative examples are provided in Appendix D.

|  | Acc ↑ | Prec ↑ | Rec ↑ | F1 ↑ | EDist ↓ |
|---|---|---|---|---|---|
| (a) VD [44] | 0.777 | 0.773 | 0.716 | 0.724 | 0.44 |
| (b) C-VAE [45] | 0.74 | 0.718 | 0.704 | 0.702 | 0.381 |
| (c) GNN [46] | 0.73 | 0.786 | 0.686 | 0.674 | 0.531 |
| (d) Ours | **0.924** | **0.914** | **0.911** | **0.912** | 0.277 |
| (e) Mesh [9] | 0.747 | 0.739 | 0.723 | 0.712 | **0.269** |

Table 1: Baseline comparison: (a) Vertex-Diffusion, (b) Conditional-VAE, (c) GNN, (d) VisDiff, (e) MeshAnything. **Acc**: Accuracy, **Prec**: Precision, **Rec**: Recall, **EDist**: Euclidean distance between point sets for triangulation evaluation.

## 6.3 Visibility Characterization

VisDiff can address the *Visibility Characterization* problem by varying the diffusion process seeds. Figure 3 illustrates how VisDiff produces distinct polygons with varying perturbations while preserving visibility consistent with the ground-truth graph $G$. Additional qualitative results are provided in Appendix E.

We introduce two metrics, coverage and diversity, to quantitatively evaluate the *Visibility Characterization* problem.

The diversity for a given visibility graph $G$ is computed by sampling $N$ valid polygons using the recognition method in Section F and calculating pairwise Chamfer distances between their point sets. The Chamfer distance captures geometric variation while remaining invariant to vertex ordering. VisDiff achieves a mean Chamfer distance of 0.56 across the test set with $N = 50$, which corresponds to roughly 20% of the 2×2 domain and indicates substantial diversity among the generated polygons.

The coverage metric measures the extent of latent space exploration for a given visibility graph $G$. We propose a metric-based exploration algorithm to evaluate this coverage. We initialize the root polygon $P_0$ by sampling a latent noise vector with base standard deviation $\sigma$ and generating a polygon using VisDiff. A breadth-first exploration is then performed up to a fixed depth $d$ and branching factor $b$, where each child is generated by adding scheduled noise to its parent in latent space. A node is expanded only if (i) the generated polygon is valid with an F1-score above threshold $T$ using the recognition method in Section F, and (ii) it is distinct from previously discovered nodes based on distance threshold $T_d$. The coverage metric is defined as the ratio of expanded nodes to the maximum possible nodes, representing the fraction of the latent neighborhood explored. Table 2 reports the performance of VisDiff across different hyperparameters. On average, VisDiff expands about 51% of possible nodes across the test set, given 20 training augmentations per visibility graph. This demonstrates that the latent exploration strategy effectively discovers a broader set of valid solutions than those encountered during training.

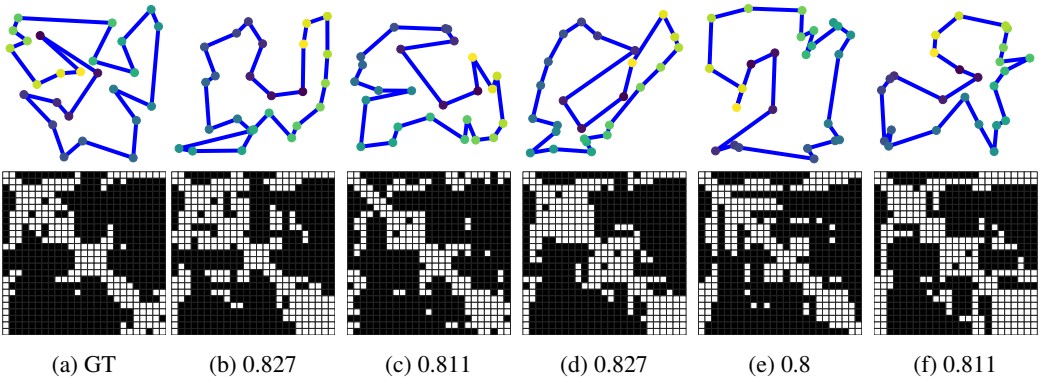

| (a) GT | (b) 0.827 | (c) 0.811 | (d) 0.827 | (e) 0.8 | (f) 0.811 |

Figure 3: *Visibility Characterization*: The top row shows multiple polygons generated by VisDiff for the same visibility graph $G$. The first vertex is represented by deep purple and the last vertex by yellow (anticlockwise ordering). The second row shows the visibility graph corresponding to the polygons where **white** denote visible edge and **black** denote non-visible edge. The caption shows the F1-Score compared to the ground truth (GT) visibility graph.

| F1 Threshold $T$ | Depth $d$ | Branching Factor $b$ | Distance Threshold $T_d$ | Coverage Metric ↑ |
|---|---|---|---|---|
| 0.85 | 5 | 2 | 0.1 | 0.475 |
| 0.80 | 5 | 2 | 0.1 | 0.488 |
| 0.75 | 5 | 2 | 0.1 | 0.495 |
| 0.70 | 5 | 2 | 0.1 | **0.515** |

Table 2: Coverage metric for different hyperparameters. Higher coverage indicates broader exploration of the latent space.

## 6.4 Visibility Recognition

We present preliminary results on the *Visibility Recognition* problem. We generate a test set of 50 valid and non-valid visibility graphs for this task. Polygons with holes are used as examples of non-valid visibility graphs. A polygon with a hole has both an outer boundary and one or more inner boundaries, making it a non-simple polygon. The visibility graph is computed in the same way as for

simple polygons. However, any edge passing through a hole is considered non-visible, as the hole region lies outside the polygon.

To determine whether a given visibility graph $G$ is valid, we first sample a set of polygons $S$ from $G$ using VisDiff. $G$ is classified as a valid graph if any polygon in $S$ is valid and achieves an F1-score above a predefined threshold $X$. Figure 4b presents qualitative results of polygon generation by VisDiff for a non-valid visibility graph. In this case, VisDiff fails to generate any valid polygon with an F1-score above 0.85, leading to its classification as a non-valid visibility graph. Figure 4c shows the performance of our model on the *Visibility Recognition* problem under different F1-score thresholds. VisDiff correctly classifies 90% of the samples from the set of valid and non-valid visibility graphs when the F1 threshold is set near the mean performance observed on the *Visibility Reconstruction* problem. This classification accuracy demonstrates that VisDiff effectively captures and represents the underlying space of valid visibility graphs. Additional qualitative results for the *Visibility Recognition* problem are provided in Appendix F.

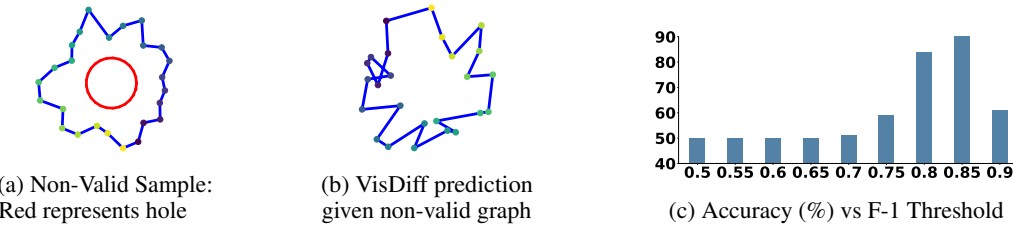

(a) Non-Valid Sample: Red represents hole

(b) VisDiff prediction given non-valid graph

(c) Accuracy (%) vs F-1 Threshold

Figure 4: We provide qualitative and quantitative results of VisDiff on *Visibility Recognition* problem.

## 6.5 Triangulation

In this section, we replace the visibility graph input with the triangulation graph to demonstrate the versatility of VisDiff for the reconstruction problem. Note that a polygon may admit multiple valid triangulations, each consisting of $n - 2$ triangles, where $n$ is the number of vertices [47]. We employ the Constrained Delaunay Triangulation [48] to triangulate the polygons in our dataset, ensuring a unique triangulation for each polygon [49]. To increase data diversity, we perform rotation-based augmentation to generate multiple samples per triangulation graph. Rotation, rather than shear augmentation, is used because the triangulation graph structure changes with perturbations in vertex locations.

We evaluate the performance on the triangulation reconstruction problem by computing the Euclidean distance between corresponding vertices, as each triangulation is uniquely determined by the spatial configuration of the points. To account for rotation variations, all polygons are rotated such that their first edge is aligned with the x-axis. Table 1 presents the quantitative results of VisDiff compared to the baselines on the triangulation graph reconstruction problem. VisDiff performs comparably to MeshAnything and outperforms Vertex-Diffusion, GNN, and Conditional-VAE in terms of Euclidean distance. Since MeshAnything is trained using a triangulation-based representation, its strong performance on triangulation structures is expected. However, it is important to note that MeshAnything frequently produces disconnected triangle sets, whereas our approach consistently generates a closed polygonal chain. Metrics for MeshAnything are reported only for polygons forming a closed polygonal chain. Qualitative results are provided in Appendix D.4.

## 6.6 Polygon Sampling

In previous experiments, we demonstrated that VisDiff can learn to generate multiple polygons conditioned on the visibility and triangulation graphs. In this section, we leverage these capabilities to highlight the high-diversity sampling ability of VisDiff over the space of all polygons. Specifically, we apply VisDiff to generate valid interpolations between pairs of polygons that share the same visibility graph. Beyond polygon-to-polygon interpolation, we also introduce a graph-to-graph interpolation approach to sample diverse polygons across the polygon manifold.

**Polygon-to-Polygon Interpolation:** We perform polygon-to-polygon interpolation by sampling two different diffusion seeds and linearly interpolating between them. VisDiff is then used to generate polygons corresponding to the interpolated noise samples while keeping the visibility graph

fixed. We perform 50 interpolation steps in total. Qualitative results are provided in Appendix G.1, showing six intermediate steps from the polygon-to-polygon interpolation process. VisDiff produces meaningful intermediate polygons across the interpolation sequence, indicating that it learns a smooth neighborhood structure within the latent space of a visibility graph.

**Graph-to-Graph Interpolation:** Sampling between two valid graphs through interpolation is a challenging problem, as intermediate steps must correspond to valid graphs for which polygons exist. Triangulation graphs possess an important property: any two valid triangulations can be transformed into one another through a sequence of local operations known as *edge flips*, with all intermediate triangulations remaining valid [50]. This sequence forms a *flip graph*, which has been shown to be connected for 2D point sets [51]. We therefore exploit the capability of VisDiff to sample using intermediate triangulation graphs, enabling graph-to-graph interpolation over valid polygonal structures.

We first select two distinct valid triangulation graphs from the test dataset and apply the edge-flip algorithm [50] to generate intermediate triangulations between them. These intermediate triangulation paths are then used as inputs to VisDiff to generate the corresponding polygons. Qualitative results in Appendix G.2 show that VisDiff produces smooth interpolations between triangulation graphs. This observation further indicates that VisDiff learns a continuous manifold representation for triangulation graphs.

## 7 Applications

VisDiff enables several practical applications. The proposed dataset serves as a navigation benchmark for evaluating motion planners in high-occlusion settings. An example of this application is presented in [52]. Its polygon sampling capability supports data augmentation by generating diverse, valid polygons conditioned on the underlying graph structure. Finally, the insights from VisDiff can be integrated into floorplan generative models [53, 54] to incorporate privacy constraints in generated layouts.

## 8 Limitations and Future Work

At a high level, our results demonstrate that modern neural representations can encode the space of all polygons such that distances on the learned manifold remain faithful to their combinatorial properties. However, the current VisDiff architecture represents the SDF as a grid, leading to computational and memory bottlenecks as polygons with more vertices require finer grid resolutions. We plan to adopt more efficient SDF encodings, such as [55, 56]. Another limitation of VisDiff is that it does not achieve perfect visibility graph reconstruction accuracy. To address this, we aim to explore LLM-based coding approaches [57, 58] to incorporate additional consistency mechanisms.

## 9 Conclusion

In this paper, we studied the problems of reconstruction and characterization of simple polygons based on combinatorial graphs that lack an underlying distance metric, features, or neighborhood structure. We introduced **VisDiff**, a diffusion-based approach that first predicts the Signed Distance Function (SDF) associated with the input visibility graph $G$. The SDF is then used to generate the vertex locations of a polygon $P$ whose visibility graph corresponds to $G$. We showed that incorporating the SDF leads to a 26% improvement in F1-score compared to state-of-the-art approaches on the *Visibility Reconstruction* problem. We further demonstrated the ability of VisDiff to sample multiple polygons for a single visibility graph $G$, addressing the *Visibility Characterization* problem. Additionally, we presented preliminary results achieving 90% classification accuracy on the *Visibility Recognition* problem. Beyond visibility graphs, VisDiff also generalizes to triangulation as inputs. Finally, we leveraged these capabilities to perform diverse polygon sampling through both graph-to-graph and polygon-to-polygon interpolation.

**Acknowledgment.** This work was funded in part by the National Research Foundation of Korea (NRF) grant (MSIT) No. RS-2024-00462874.

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

# Appendix

We first provide formal definitions of the terms used in this paper in Section A. Next, we describe the dataset generation process in Section B. The ablation study analyzing the design choices of our method is presented in Section C. We then provide additional results for *Visibility Reconstruction* (Section D), *Visibility Characterization* (Section E), and *Visibility Recognition* (Section F). In Section G, we present further results demonstrating the ability of VisDiff to sample polygons on a continuous polygon manifold. We also evaluate intermediate results of the SDF diffusion process in Section H and analyze the effect of SDF error on the vertex generation block in Section I. Finally, we provide model hyperparameters, architectural details, and training configurations in Section J.

## A    Definitions

We provide more formal definitions of the terms simple polygon and visibility graph in Table 3.

| Terms | Definitions |
|-------|-------------|
| Simple Polygon | Let $V = (v_1, \ldots, v_n)$ be an ordered set of $n$ points on the plane. The location of point $v_i$ is specified by its coordinates $(x_i, y_i)$. Let $e_i = (v_i, v_{i+1})$ be the set of line segments obtained by connecting consecutive points in $V$ in a cyclic manner. These line segments define a closed planar curve – the boundary of a polygon $P$. The points $v_i$ are the *vertices* of $P$ and the segments $e_i$ are its *sides*. 

 Two consecutive edges of a polygon share an end-point at a vertex. In a simple polygon, these are the only intersections between the edges. The edges do not intersect each other. |
| Visibility Graph | A simple polygon $P$ has a well-defined interior and an exterior separated by its boundary $\delta P$. This separation allows us to define *visibility*: We will use the notation $x \in P$ to denote that $x$ lies either on the boundary or the interior of $P$. We say that two points $x, y \in P$ *see each other* if and only if $\forall z \in [x, y], z \in P$. In other words, the line segment $[xy]$ lies completely inside or on the boundary of $P$. 

 The visibility graph of $P$, denoted $G(P)$, is a graph that is a vertex-to-vertex relation of $P$. There is an edge between two vertices $u$ and $v$ if and only if $u$ and $v$ are visible to each other in $P$. |

Table 3: Definitions

## B    Dataset Generation

### B.1    Dataset Statistics

In this section, we present detailed statistics of our dataset. Figure 5 shows the distribution of the training and in-distribution test set, indicating that our dataset is uniform with respect to the link diameter of the visibility graph. Figure 6 compares the training dataset with the out-of-distribution test set, showing that the star, convex-fan, and terrain classes have edge densities that differ from the training distribution, where density refers to the ratio of visible edges to total possible edges in the visibility graph. Table 4 provides a comparison between the VisDiff dataset and other real-world datasets in terms of visibility graph diversity based on link diameter. The results demonstrate that the VisDiff dataset exhibits significantly greater diversity than existing real-world datasets.

### B.2    Test Set Generation

We generate two datasets for evaluation: in-distribution and out-of-distribution. In-distribution samples are generated by setting aside 100 unique polygons per link diameter from the large dataset. These are not included in the training data.

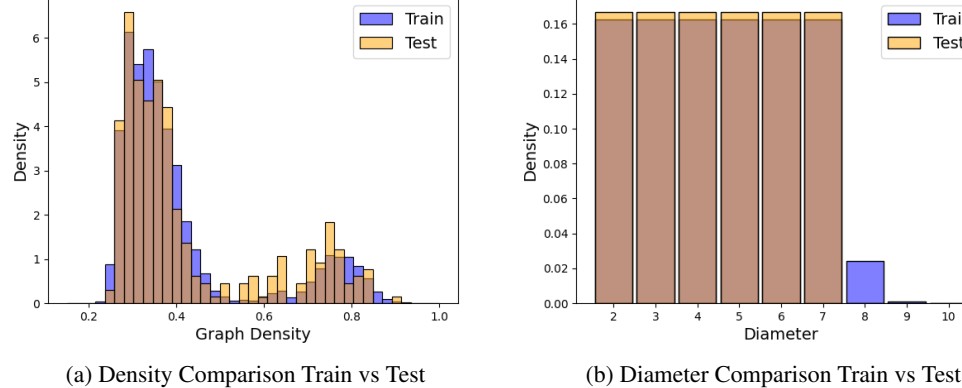

(a) Density Comparison Train vs Test  (b) Diameter Comparison Train vs Test

Figure 5: Train vs in-distribution test set analysis: 5a) The density is inversely proportional to the diameter. Uniform sampling of diameter results in bimodal density. 5b) Training and testing sets are uniform in terms of the link diameter of the visibility graph.

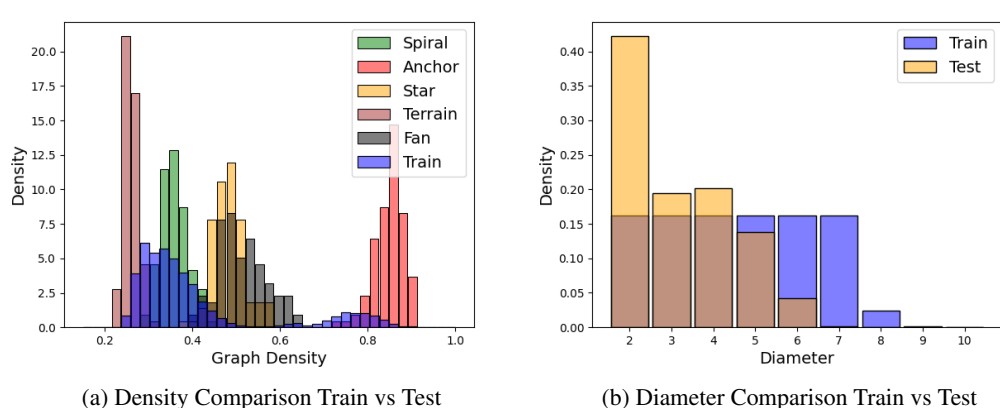

(a) Density Comparison Train vs Test  (b) Diameter Comparison Train vs Test

Figure 6: Out-of-distribution test set analysis: Figure 6a shows the density of the anchor and spiral are close to the mean of the bimodal training distribution, making it similar to our training set. The density of the star, convex fan, and terrain differ significantly from the training distribution.

| Baselines | Link Diameter (Avg / Std) | Min | Max |
|---|---|---|---|
| MNIST [59] | 1.83 / 0.50 | 1.0 | 4.1 |
| COCO 2017 [60] | 1.32 / 0.25 | 1.0 | 3.823 |
| VisDiff | **4.4 / 2.2** | 1.0 | **9.0** |

Table 4: Real-world dataset comparison showing the diversity (in link diameter) between the VisDiff dataset and standard benchmarks.

The out-of-distribution samples are generated based on specific polygon types: star, spiral, anchor, convex-fan, and terrain. Figure 7 illustrates the properties of these polygon types. The spiral and anchor polygons share characteristics similar to those in our dataset, whereas the terrain, convex-fan, and star polygons differ significantly in terms of density, defined as the ratio of visible edges to total possible edges in the visibility graph. Figure 6 highlights the differences in visibility graph density for the terrain, convex-fan, and star classes compared to the training set.

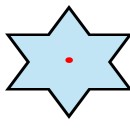 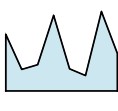 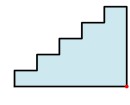 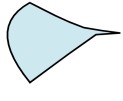 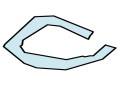

(a) Star       (b) Terrain       (c) Fan       (d) Anchor       (e) Spiral

Figure 7: Polygon types: a) **Star**: Single kernel point (red) from which all vertex locations are visible, b) **Terrain**: X-monotone polygons where orthogonal lines from the X axis intersect the polygon boundary at most twice, c) **Convex Fan**: Single convex vertex (red) which appears in every triangle of the polygon triangulation, d) **Anchor**: Polygons with two reflex links and a convex link connecting both of them, e) **Spiral**: Polygons with long link diameter.

## C   Ablation Study

In this section, we present a performance comparison of our architecture using different feature types. Specifically, we analyze the contribution of each feature type to visibility reconstruction performance through an ablation study. Table 5 summarizes the results of this comparison.

|  | Acc ↑ | Prec ↑ | Rec ↑ | F1 ↑ |
|---|---|---|---|---|
| (a) Global | 0.894 | 0.884 | 0.873 | 0.876 |
| (b) Pixel | 0.869 | 0.859 | 0.839 | 0.843 |
| (c) VisDiff | **0.924** | **0.914** | **0.911** | **0.912** |

Table 5: Ablation comparison: (a) Global patch-based features, (b) Pixel-aligned local features, and (c) Combined global patch-based + pixel-aligned local features. **Acc**: Accuracy, **Prec**: Precision, **Rec**: Recall.

## D   Visibility Reconstruction Results

In this section, we present a quantitative comparison of VisDiff with baselines on the out-of-distribution dataset for the *Visibility Reconstruction* problem. We also provide additional qualitative results for the *Visibility Reconstruction* problem.

### D.1   Out-of-distribution Comparison

In this section, we present the baseline performance on the out-of-distribution dataset for the *Visibility Reconstruction* problem. Table 6 compares VisDiff with baseline methods on this dataset. The F1-score comparison shows that VisDiff performs significantly better than all baselines on the out-of-distribution dataset.

|  | Acc ↑ | Prec ↑ | Rec ↑ | F1 ↑ |
|---|---|---|---|---|
| (a) VD | 0.751 | 0.734 | 0.702 | 0.697 |
| (b) C-VAE | 0.733 | 0.713 | 0.699 | 0.694 |
| (c) GNN | 0.666 | 0.732 | 0.643 | 0.610 |
| (d) Ours | **0.915** | **0.895** | **0.891** | **0.891** |
| (e) Mesh | 0.708 | 0.715 | 0.709 | 0.675 |

Table 6: Baseline comparison: (a) Vertex-Diffusion, (b) Conditional-VAE, (c) GNN, (d) VisDiff, and (e) MeshAnything. **Acc**: Accuracy, **Prec**: Precision, **Rec**: Recall.

### D.2   Reconstruction Generation Strategy

In this section, we describe the approach used in VisDiff for polygon generation and selection in the *Visibility Reconstruction* problem. We leverage the capability of VisDiff to generate multiple polygon

samples for a given visibility graph. The visibility graph of each generated polygon is computed, and the final polygon is selected based on the highest F1-score with respect to the target visibility graph. We use 50 samples, chosen as a trade-off between performance and computational cost across different sample sizes. For a fair comparison, we apply the same generation and selection procedure to Vertex Diffusion and C-VAE, since both methods are also capable of generating multiple polygons for a given visibility graph.

## D.3   Qualitative Results

We provide additional qualitative results for the *Visibility Reconstruction* problem. Figures 8, 9 and 10 show the comparison between polygons generated by VisDiff to baselines. The F1-Score shows that VisDiff generates polygons much closer to the visibility graph of the ground truth polygon.

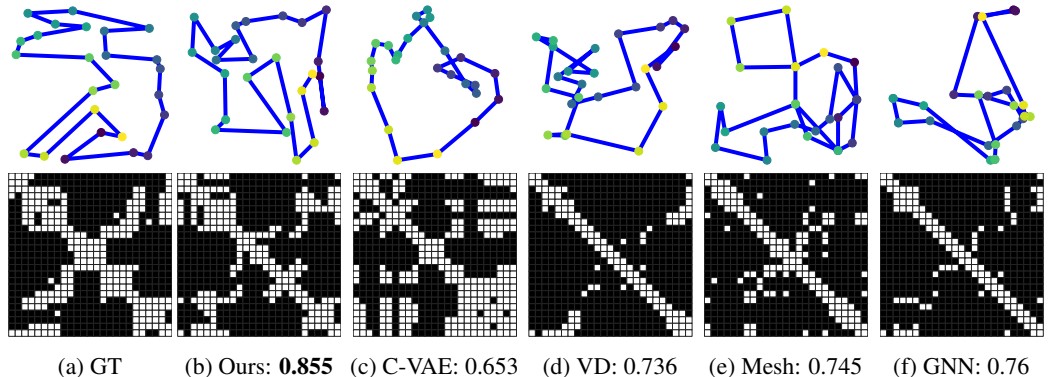

(a) GT        (b) Ours: **0.855**   (c) C-VAE: 0.653   (d) VD: 0.736   (e) Mesh: 0.745   (f) GNN: 0.76

Figure 8: *Visibility Reconstruction*: The top row shows the polygons generated by different methods. The first vertex is represented by deep purple and the last vertex by yellow (anticlockwise ordering). The second row shows corresponding visibility graphs of the polygons where **white** represents the visible edge and **black** represents the non-visible edge. The captions indicate the F1 Score of the visibility graph compared to the GT.

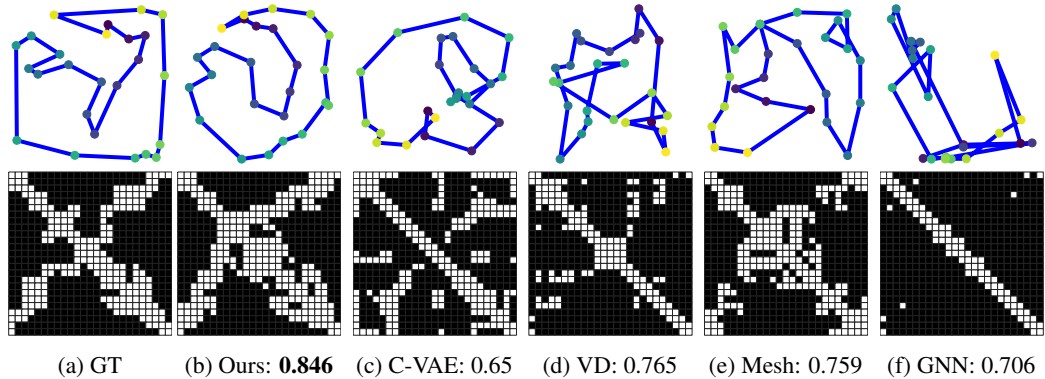

(a) GT        (b) Ours: **0.846**   (c) C-VAE: 0.65   (d) VD: 0.765   (e) Mesh: 0.759   (f) GNN: 0.706

Figure 9: Visibility reconstruction qualitative results: The top row shows the polygons generated by different methods. The first vertex is represented by deep purple and the last vertex by yellow (anticlockwise ordering). The second row shows corresponding visibility graphs of the polygons where **white** represents the visible edge and **black** represents the non-visible edge. The captions indicate the F1 Score of the visibility graph compared to the GT.

## D.4   Triangulation Results

In addition to conditioning on the visibility graph, we also provide qualitative results for the problem of generating polygons from triangulation graphs. Figures 11 and 12 show the performance of VisDiff compared to other baselines. It can be observed that both MeshAnything and VisDiff generate

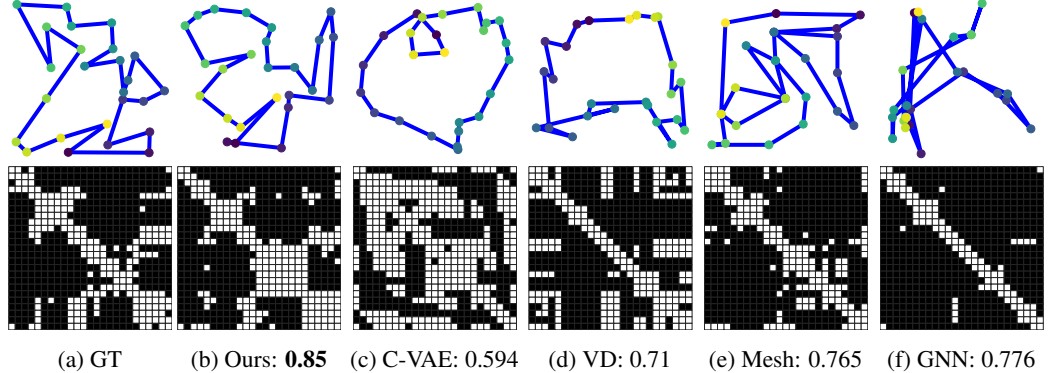

| (a) GT | (b) Ours: **0.85** | (c) C-VAE: 0.594 | (d) VD: 0.71 | (e) Mesh: 0.765 | (f) GNN: 0.776 |

Figure 10: Visibility reconstruction qualitative results: The top row shows the polygons generated by different methods. The first vertex is represented by deep purple and the last vertex by yellow (anticlockwise ordering). The second row shows corresponding visibility graphs of the polygons where **white** represents the visible edge and **black** represents the non-visible edge. The captions indicate the F1 Score of the visibility graph compared to the GT.

polygons close to the ground truth, whereas Vertex Diffusion, C-VAE, and GNN fail to accurately interpret the triangulation graph.

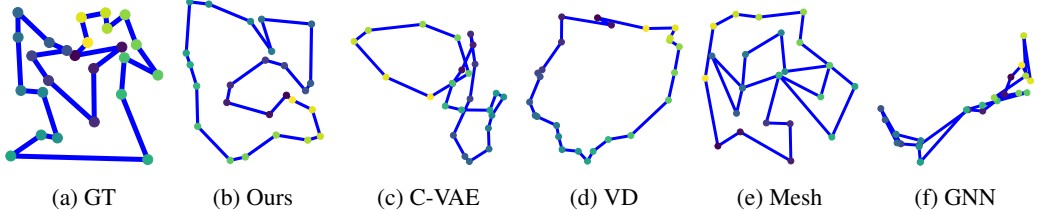

| (a) GT | (b) Ours | (c) C-VAE | (d) VD | (e) Mesh | (f) GNN |

Figure 11: Triangulation Qualitative Results: Top row shows the polygons generated by different methods. The first vertex is represented by deep purple and the last vertex by yellow (anticlockwise ordering).

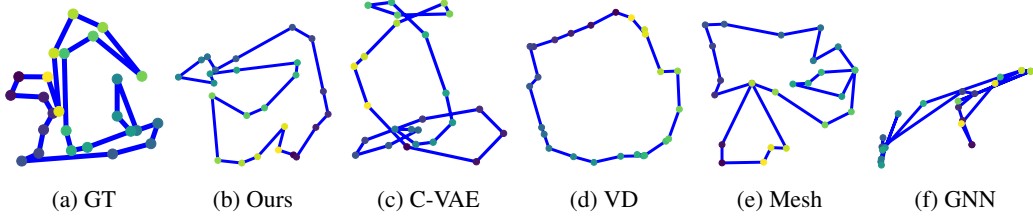

| (a) GT | (b) Ours | (c) C-VAE | (d) VD | (e) Mesh | (f) GNN |

Figure 12: Triangulation Qualitative Results: Top row shows the polygons generated by different methods. The first vertex is represented by deep purple and the last vertex by yellow (anticlockwise ordering).

## D.5 Computational Cost Comparison

In Table 7, we compare the computational cost of our model with the baselines by measuring the inference time per sample. The inference time of VisDiff is higher than that of the baseline models, as VisDiff performs inference in two stages through the SDF, whereas the baselines complete it in a single step.

| Baselines | Computational Time (seconds) ↓ |
|---|---|
| Mesh | 0.40 |
| C-VAE | 0.003 |
| GNN | 0.005 |
| VD | 0.094 |
| Ours | 1.02 |

Table 7: Computational cost comparison: Inference time (in seconds) required by each model to generate vertex locations for a single visibility graph.

# E  Visibility Characterization Results

We provide additional qualitative results for the *Visibility Characterization* problem. Figures 13 and 14 demonstrate the ability of VisDiff to sample multiple polygons given the same visibility graph.

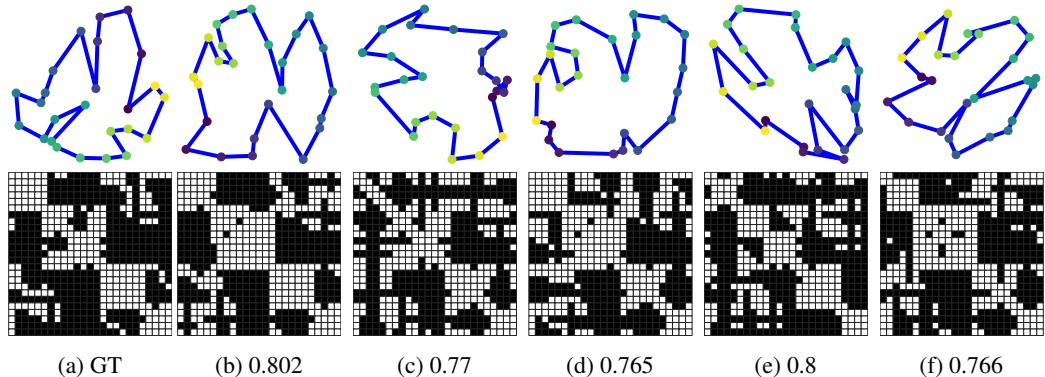

| (a) GT | (b) 0.802 | (c) 0.77 | (d) 0.765 | (e) 0.8 | (f) 0.766 |

Figure 13: *Visibility Characterization*: The top row shows multiple polygons generated by VisDiff for the same visibility graph $G$. The first vertex is represented by deep purple and the last vertex by yellow (anticlockwise ordering). The second row shows the visibility graph corresponding to the polygons where **white** represents visible edge and **black** represents non-visible edge. The caption shows the F1-Score compared to the ground truth (GT) visibility graph.

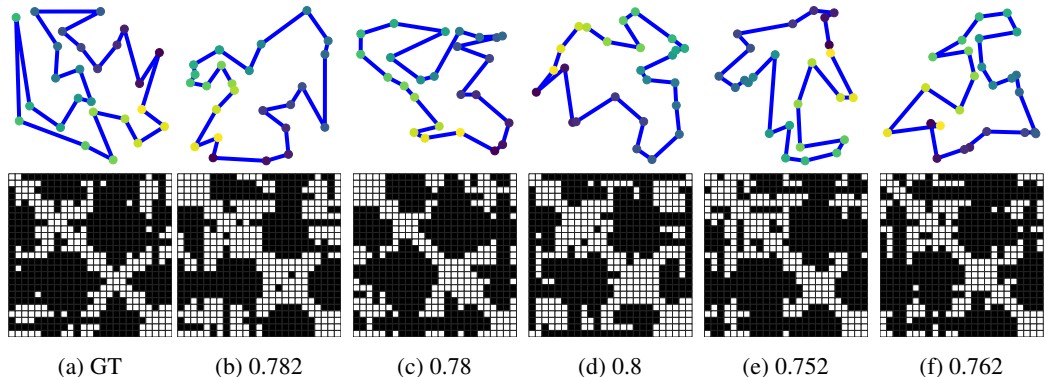

| (a) GT | (b) 0.782 | (c) 0.78 | (d) 0.8 | (e) 0.752 | (f) 0.762 |

Figure 14: *Visibility Characterization*: The top row shows multiple polygons generated by VisDiff for the same visibility graph $G$. The first vertex is represented by deep purple and the last vertex by yellow (anticlockwise ordering). The second row shows the visibility graph corresponding to the polygons where **white** represents visible edge and **black** represents non-visible edge. The caption shows the F1-Score compared to the ground truth (GT) visibility graph.

# F  Visibility Recognition Results

We provide additional qualitative results illustrating both successful and failure cases of VisDiff on the *Visibility Recognition* problem, where the input may be a valid or non-valid visibility graph. Invalid samples are generated by constructing visibility graphs of polygons with holes. Figures 15a and 15c show successful reconstructions for valid and non-valid visibility graphs, respectively, while Figures 15b and 15d illustrate failure cases. These results demonstrate that VisDiff can identify non-valid visibility graphs in most scenarios when used as a classifier based on the validity of its generated outputs.

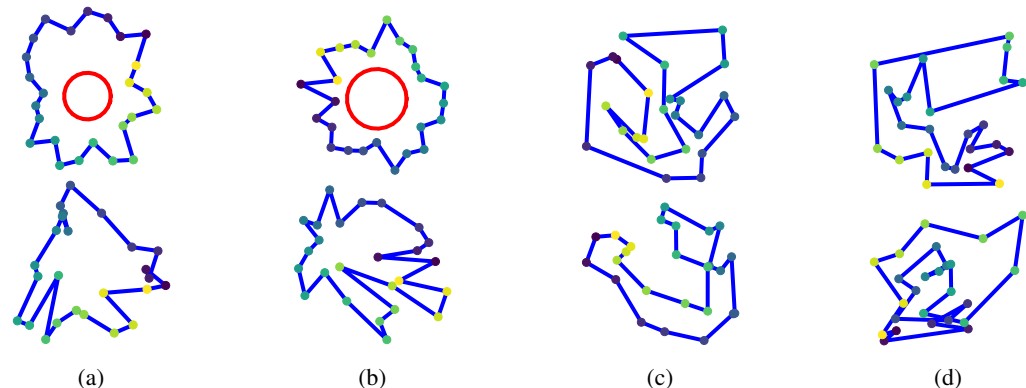

    (a)             (b)             (c)             (d)

Figure 15: *Visibility Recognition*: The top row signifies the ground truth non-valid polygon with the hole (red) while the bottom row is the polygons drawn by VisDiff. The first vertex is represented by deep purple and the last vertex by yellow (anticlockwise ordering). a) Non-Valid Sample 1: VisDiff predicts it as a non-valid polygon as it is not able to generate any valid polygon, b) Non-Valid Sample 2: VisDiff generates valid polygon where it learns to put points in a $V$ shape to account for a hole. It misclassified a non-valid visibility graph as a valid visibility graph. c) Valid Sample 1: VisDiff predicts it as a non-valid polygon as it is not able to generate any valid polygon, d) Valid Sample 2: VisDiff predicts it as a valid visibility graph as it is able to generate any valid polygon with high F1 relative to target visibility graph

# G  Polygon Sampling

We provide qualitative results demonstrating the capability of VisDiff to perform high-diversity data sampling within the polygon manifold space. We present qualitative examples for both the polygon-to-polygon interpolation and graph-to-graph interpolation approaches.

## G.1  Polygon-to-Polygon Interpolation

We provide qualitative results for the polygon-to-polygon interpolation approach. Figures 16 and 17 visualize six interpolation steps for two polygon samples. VisDiff generates meaningful intermediate polygons across the interpolation sequence.

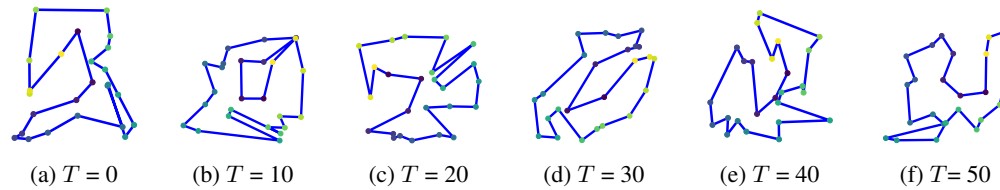

  (a) $T = 0$     (b) $T = 10$     (c) $T = 20$     (d) $T = 30$     (e) $T = 40$     (f) $T = 50$

Figure 16: Polygon-to-Polygon Interpolation: The top row shows different polygons generated by VisDiff at different instances of 50 steps during linear interpolating between two noise samples the same visibility graph $G$. The first vertex is represented by deep purple and the last vertex by yellow (anticlockwise ordering). The caption shows the timestep of interpolation between the two samples.

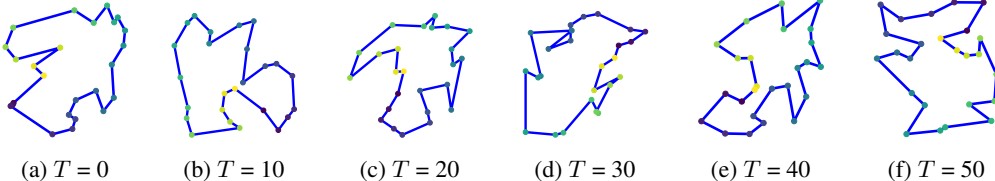

| (a) $T = 0$ | (b) $T = 10$ | (c) $T = 20$ | (d) $T = 30$ | (e) $T = 40$ | (f) $T = 50$ |

Figure 17: Polygon-to-Polygon Interpolation: The top row shows different polygons generated by VisDiff at different instances of 50 steps during linear interpolating between two noise samples the same visibility graph $G$. The first vertex is represented by deep purple and the last vertex by yellow (anticlockwise ordering). The caption shows the timestep of interpolation between the two samples.

## G.2 Graph-to-Graph Interpolation

We provide qualitative results for the graph-to-graph interpolation approach. Figures 18 and 19 visualize six randomly sampled intermediate interpolation steps for two triangulation graph examples. VisDiff generates meaningful intermediate polygons between the triangulation graphs of convex and concave polygons.

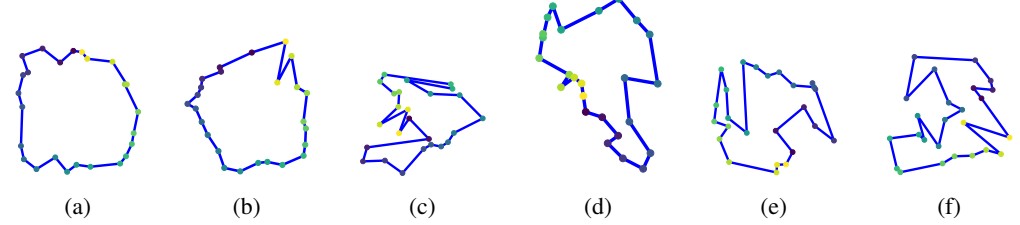

| (a) | (b) | (c) | (d) | (e) | (f) |

Figure 18: Graph-to-Graph Interpolation: The top row shows different polygons generated by VisDiff at different interpolation instances of two triangulation graphs. The first vertex is represented by deep purple and the last vertex by yellow (anticlockwise ordering). The caption shows the timestep of interpolation between the two samples.

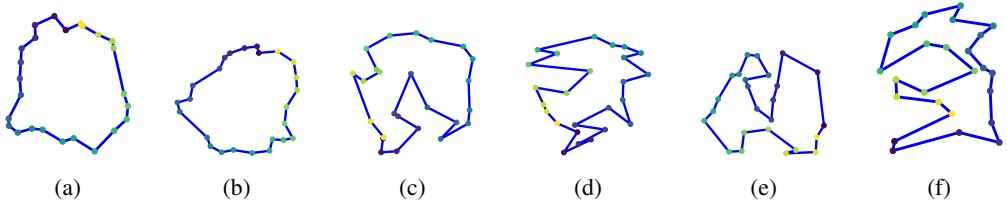

| (a) | (b) | (c) | (d) | (e) | (f) |

Figure 19: Graph-to-Graph Interpolation: The top row shows different polygons generated by VisDiff at different interpolation instances of two triangulation graphs. The first vertex is represented by deep purple and the last vertex by yellow (anticlockwise ordering). The caption shows the timestep of interpolation between the two samples.

## H SDF Diffusion Evaluation

We evaluate the SDF diffusion model by measuring the L2 error between the ground-truth and predicted SDFs on both in-distribution and out-of-distribution test datasets for the *Visibility Reconstruction* problem. Table 8 presents the performance of the SDF diffusion model. The results show that our diffusion model predicts high-quality SDFs with low L2 error, demonstrating its effectiveness in capturing the underlying relationship between polygons and their visibility graphs.

| Test Dataset | L2 Error ↓ |
|---|---|
| In-Distribution | 0.071 |
| Out-Distribution: Spiral | 0.091 |
| Out-Distribution: Terrain | 0.091 |
| Out-Distribution: Convex Fan | 0.083 |
| Out-Distribution: Anchor | 0.158 |
| Out-Distribution: Star | 0.069 |

Table 8: SDF evaluation: L2 error between the predicted SDF from the diffusion model and the ground-truth SDF across in- and out-distribution test sets.

# I    Vertex Prediction Analysis

In this section, we present experiments analyzing the effect of SDF generation on the vertex prediction block. We also provide a quantitative comparison between traditional polygon extraction methods and our vertex prediction block.

## I.1    SDF-to-Polygon Error Analysis

We analyze the impact of perturbations in the SDF on the downstream vertex extraction block. Specifically, we report changes in performance metrics when varying levels of noise are added to the SDF input. We also evaluate the effect on the vertex prediction block when the visibility conditioning is removed during SDF generation. Table 9 summarizes the performance of the vertex prediction block across both experiments. As the SDF becomes increasingly distorted, the accuracy of vertex extraction consistently degrades, since the zero-level set becomes harder to localize and fails to form a well-defined polygon consistent with the visibility graph. We also observe a noticeable drop in performance compared to conditional SDF and polygon generation, indicating that reconstruction quality strongly depends on the fidelity of the conditional SDF.

| SDF Noise Standard Deviation | F1 Score ↑ |
|---|---|
| 0.0 (Conditional SDF) | **0.912** |
| 0.0 (Unconditional SDF) | 0.865 |
| 0.01 | **0.912** |
| 0.05 | 0.909 |
| 0.1 | 0.905 |
| 0.5 | 0.859 |

Table 9: SDF-to-polygon error analysis: F1 score between the visibility graph of the generated polygon (from the vertex prediction block) and the ground-truth polygon under varying SDF noise levels. Conditional and unconditional SDF generation results are also compared.

## I.2    Vertex Prediction Baseline Comparison

We compare our proposed vertex prediction block with the standard polygon extraction approach, specifically the Marching Cubes algorithm. Table 10 presents a comparison between VisDiff and Marching Cubes. VisDiff significantly outperforms Marching Cubes across all evaluation metrics.

| Approach | F1 Score ↑ |
|---|---|
| VisDiff | **0.912** |
| Marching Cubes Algorithm [61] | 0.800 |

Table 10: Vertex prediction baseline comparison: F1 score between the visibility graph of the polygon generated by the vertex prediction block and the ground-truth polygon, compared against a traditional polygon extraction approach (Marching Cubes).

# J    Training Details

In this section, we present the training setup and detailed architecture of **VisDiff**. We also provide a quantitative comparison with the joint training variant of VisDiff. All models were trained on a single NVIDIA A100 GPU using 10 workers, with a total training time of approximately 16 hours.

## J.1    SDF Diffusion

The SDF diffusion block uses a time-conditioned U-Net architecture with three downsampling layers having channel sizes of 32, 64, and 128. Each downsampling block is followed by a Spatial Transformer layer, which performs cross-attention with the visibility features. The bottleneck feature consists of 512 channels and is followed by upsampling layers with channel sizes of 128, 64, 32, and finally 1. Skip connections are maintained between the encoder and decoder, consistent with the standard U-Net structure. Additionally, a Spatial Transformer layer is applied after each upsampling block, and all U-Net layers use ReLU activation functions.

We train the SDF diffusion model for 60 epochs using the Adam optimizer with a learning rate of $10^{-4}$ and a batch size of 128. We employ a log-linear noise scheduler with $\sigma_{\min} = 0.005$ and $\sigma_{\max} = 10$.

## J.2    Vertex Extraction Block

The vertex extraction block consists of two modules: the SDF encoding module and the vertex prediction block. Both modules were trained for 60 epochs using the Adam optimizer with a learning rate of $10^{-4}$ and a batch size of 128. We now describe the architecture and training setup used for these modules.

### J.2.1    SDF Encoding

The SDF encoding block uses a U-Net architecture to extract pixel-aligned features. The encoder consists of convolutional layers with channel sizes of 64, 128, 256, and 512, while the decoder contains upsampling layers with channel sizes of 256, 128, 64, and 128, producing the final pixel-aligned feature maps. All layers use ReLU activation functions. The bottleneck feature is represented as $Z_{\text{global}} \in \mathbb{R}^{5 \times 5 \times 512}$, which is flattened to $\mathbb{R}^{25 \times 512}$. Positional embeddings are added to these flattened patches to preserve spatial ordering.

### J.2.2    Vertex Prediction Block

The vertex prediction block consists of three transformer encoder layers, each with 256 hidden units. The contour initialized from the SDF is provided as input to the transformer, and the output passes through a multilayer perceptron (MLP) with layers of 256 and 2 units to predict the final $(x, y)$ coordinates for each polygon query. Figure 20 illustrates the detailed architecture of the transformer block.

## J.3    Joint vs. Two-Stage Training

We compare the performance of our two-stage training approach with joint training of the SDF diffusion and vertex prediction blocks without vertex initialization. Table 11 presents the results of this comparison. Our two-stage training with vertex initialization significantly outperforms the joint training approach.

| Training Approach | F1 Score ↑ |
|---|---|
| Joint Training | 0.850 |
| Two-Stage Training with Vertex Initialization | **0.912** |

Table 11: Training approach comparison: F1 score between the visibility graph of the polygon generated by the vertex prediction block and the ground-truth polygon, comparing joint training and two-stage training with vertex initialization.

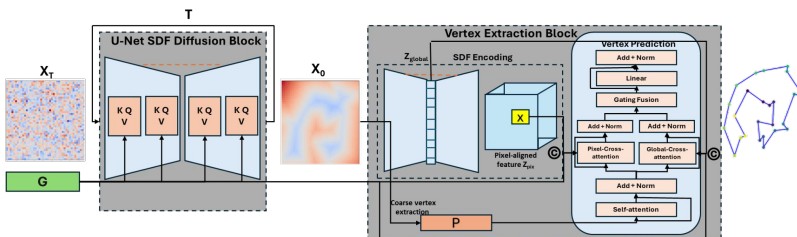

Figure 20: VisDiff architecture. The model consists of two main components: the U-Net SDF Diffusion block and the Vertex Extraction block. **U-Net Diffusion Block:** A noisy SDF, denoted as $\mathbf{X}_T$, is first sampled from a Gaussian distribution. $\mathbf{X}_T$ then passes through $\mathbf{T}$ timesteps of the reverse diffusion process to produce the clean SDF $\mathbf{X}_0$. This denoising process is conditioned on the input graph $\mathbf{G}$ using transformer cross-attention blocks represented by $\mathbf{K}$, $\mathbf{Q}$, and $\mathbf{V}$, which correspond to the key, query, and value terms, respectively. In our approach, $\mathbf{Q}$ is obtained from the learned spatial CNN features, while $\mathbf{K}$ and $\mathbf{V}$ are derived from $\mathbf{G}$. An initial set of vertices $\mathbf{P}$ is then estimated from $\mathbf{X}_0$ through contour extraction. **Vertex Extraction Block:** Given the predicted SDF $\mathbf{X}_0$, the SDF encoder generates pixel-aligned features $Z_{pix}$ and global features $Z_{global}$. These features, together with the initial vertices $\mathbf{P}$, are passed to the vertex prediction block to estimate the final vertex locations. During **Training**, the model is supervised using both the ground-truth SDF and polygon. During **Testing**, only the visibility graph $\mathbf{G}$ is provided as input.

