# OpenReview forum: "VisDiff: SDF-Guided Polygon Generation for Visibility Reconstruction, Characterization and Recognition"
_NeurIPS.cc/2025/Conference — NeurIPS 2025 poster_

### Official Review · Reviewer_UnHa · 2025-06-24

**Clarity:** 3
**Significance:** 3
**Originality:** 3
**Rating:** 4
**Confidence:** 2

**Summary:**

The paper develops a method for reconstructing (and generating) graphs from their visibility graph. Given a line graph in the form of a "loop" without intersection, one defines the visibility graph as follows. As vertices one takes the original vertex set and between two vertices an edge is drawn if the line segment between them is fully contained in the polygon (interior and boundary).

The authors create a novel dataset for the task and propose an architecture for solving the task and the model is evaluated on this dataset. For the dataset there are three tasks.
1) Reconstruction. Given a visibility graph  can we reconstruct a plausible loop.
2) Characterization. Can we generate all possible loops for the given visibility graph.
3) Existence. Can we determine if a loop exists for a given visibility graph.

Apart from the dataset, a model is proposed to solve such a task and evaluation shows that it works.

**Questions:**

See the strengths and weaknesses.

**Ethical Concerns:**

["NO or VERY MINOR ethics concerns only"]

**Final Justification:**

When first reading the paper the reviewer was of the opinion that the manuscript was of high quality and sound work. The active engagement in the rebuttals by the authors further strengthened that belief.

In term of contribution I also believe there be sufficient novelty and contributions for a somewhat underexplored field.
Therefore, (although I am not an expert) a weak accept would be my recommendation. I will maintain my score due to the lack of expertise, but also reading the other reviews and rebuttals I would be closer to an accept (5).

**Limitations:**

Yes

**Paper Formatting Concerns:**

The tables and section headers could benefit from another check for consistency, but this only a very minor thing and is easily addressed.

**Quality:**

3

**Strengths And Weaknesses:**

The problem the authors are solving is well motivated and has (potential) real world applications. A good description of the dataset is provided and the paper is in general well-written. Evaluation is in general good and the model is well described. The appendix provides good background material and insights for reviewers (such as this reviewer) that are unfamiliar with the subject, leading to a self-contained treatise.
The problem the authors (to the reviewers limited knowledge) are solving is novel as well as their approach, and in general it can be considered an underexplored problem.

As to the weaknesses, one remark that can be made is that there are no reported  standard deviations in the experiments. Also the question of generating datasets of such nature seem abundant to the reviewer and there should be plenty of examples to be either created or already available. For instance, the contours of MNIST images viewed as graphs are 2D examples, any closed (spherical) 3D mesh is an example in 3D and creating their visibility graph should not be hard.
Hence the question is if the authors could motivate why particular this (synthetic) dataset is chosen if examples are abundant.  As the method will most likely also have potential for 3D applications, how well does the method generalize to 3D?

As a general comment, Section 6 contains various spelling errors and inconsistencies with respect to the naming of models. Please correct this  as well as using proper capitalization in the section titles.
Tables are preferred to not have vertical lines.

---

> ### Author Rebuttal · Authors · 2025-07-29
>
> We thank the reviewer for their overall positive evaluation and helpful feedback for our work. We also appreciate their recognition of the novelty of our approach to an underexplored problem. Below, we address the reviewer’s questions and concerns about the no reported standard deviation, diversity of the dataset, 3D extension and some editing changes-
> - **Diversity of dataset**: We agree with the reviewer that there are plenty of examples like MNIST images or segmentation masks which could be used to generate a dataset to represent polygon-visibility space. However, we show that our synthetic datasets have higher structural diversity compared to MNIST (converted to polygonal contours) and COCO 2017 (object masks) with respect to the combinatorial properties of the visibility graph such as link diameter. We choose link diameter as it captures the level of concavity of the polygon.
> | Dataset    | Link Diameter (Avg / Std) | Min  | Max   |
> |------------|----------------------------|------|-------|
> | MNIST      | 1.83 / 0.50                | 1.0  | 4.1   |
> | COCO 2017  | 1.32 / 0.25                | 1.0  | 3.823 |
> | VisDiff    | **4.4 / 2.2**              | 1.0  | **9.0** |
> - **3D Results**: The reviewer brings up a valuable point regarding the generalization to 3D polygons. While our current work focuses on the 2D setting to maintain a clear and well-defined scope, applying the approach to 3D is indeed a compelling direction. Visibility in 3D is significantly more complex and often counter-intuitive. For example, even if all vertices are visible, it does not necessarily imply that the interior is visible. We consider the extension to 3D a promising direction for future work and thank the reviewer for highlighting this opportunity.
> - **Standard Deviation addition to the metrics**: We thank the reviewer for the suggestion of adding standard deviation to the metrics. We agree that this would provide additional insight. We will include it in the final camera-ready version of the paper for all the metrics tables.
> - **Paper Formatting and Editing**: We appreciate the reviewer’s attention to detail in identifying typos and inconsistencies with respect to the naming of models and section headings. We apologize for these errors and will ensure that all corrections are made in the final camera-ready version of the paper, including the suggested table formatting changes.
>
> We would like to finally thank the reviewer for their time and efforts in reviewing our work and giving constructive feedback. The reviewer’s comments on the presentation of metrics and the overall content have been valuable in improving both the clarity and quality of the paper. We will ensure to address them in the camera-ready version. If there are any additional questions or clarifications, we would be happy to address them.

---

> > ### Comment · Reviewer_UnHa · 2025-08-03
> >
> > Thank you very much for the extensive response and also the responses to the other reviewers! In particular the additional experiments are great and strengthen the work. Evaluating the method on standard datasets (even if the newly proposed dataset is superior) is good for understanding and for context. This clarifies the questions of the reviewer and I wish the authors the best of luck!

---

> ### Author Response · Authors · 2025-08-07
>
> Thank you for your encouraging comments.
> We will incorporate the additional experiments in the camera-ready version of the paper to further strengthen the work. Additionally, we will also include the standard data results in the arXiv version of the paper.
> We appreciate your time, thoughtful engagement, and support throughout the review process.

---

### Official Review · Reviewer_Z2o4 · 2025-07-01

**Clarity:** 2
**Significance:** 2
**Originality:** 3
**Rating:** 5
**Confidence:** 3

**Summary:**

This paper addresses the problems of visibility reconstruction, characterization, and recognition for polygons based on visibility graphs. The authors propose VisDiff, a diffusion-based framework that generates a Signed Distance Function (SDF) conditioned on the visibility graph, from which polygon vertices are extracted. This approach aims to capture the combinatorial structure of polygons more effectively than direct coordinate prediction. VisDiff is further extended to handle triangulation graphs and demonstrates capabilities for diverse polygon sampling. To support training and evaluation, the authors curate a new dataset of polygons annotated with visibility and triangulation graphs.

**Questions:**

- Although visibility reconstruction is theoretically interesting and challenging, the paper does not provide concrete examples demonstrating the practical value or real-world applications of the approach. Could the authors elaborate on this aspect?

- It is unclear how the triangulation graph differs fundamentally from the visibility graph in the context of VisDiff. Are there specific properties that make reconstruction from triangulation graphs more difficult, and what is the main message or contribution of the triangulation subsection?

**Ethical Concerns:**

["NO or VERY MINOR ethics concerns only"]

**Final Justification:**

The "SDF to polygon generation error propagation" results align with the natural understanding of the two-stage pipeline and would be a valuable complement to the experiments.

The new studies on the coverage and diversity are very interesting and informative. Please incorporate them concisely into the paper as a study of the characterization problem.

The rebuttal has resolved my concerns from the initial review, and I increased the score to 5.

**Limitations:**

Yes

**Quality:**

3

**Strengths And Weaknesses:**

### Strengths

1. The proposed solution is well-motivated and innovative. Using an SDF as an intermediate representation is a clever way to address the challenges of predicting coordinates directly from visibility graphs.

2. Experimental results show that VisDiff outperforms diverse baselines on visibility reconstruction tasks.

3. The authors curate a dedicated dataset to rigorously evaluate their method, and the appendix provides thorough details of the dataset generation process.

4. Both the dataset and code are released, supporting reproducibility and further research.


### Weaknesses

1. The background discussion on diffusion models omits key foundational references. Section 4.1 cites DALLE-2 for theoretical aspects, which is odd since it is primarily an application paper. Additionally, DDIM is not the work that originally proposed the diffusion model framework (e.g., the forward and reverse processes) but extends DDPMs[C] for accelerated sampling. The papers that propose the theoretical framework—[A], [B], and [C]—are all missing. I would recommend a more thorough and careful review of the relevant literature.

2. Section 3 defines Characterization as "generate all polygons given a valid visibility graph" - how do you measure the coverage? The evaluation of visibility characterization is limited to qualitative examples, making it difficult to assess correctness or diversity. Could the authors include quantitative metrics to evaluate diversity or coverage of generated polygons for a given visibility graph?

3. The paper lacks experimental analysis on how errors in the generated SDF propagate to the vertex extraction stage. Given that the main novelty of the approach lies in using the SDF as an intermediate representation, it would be important to examine this aspect. For instance, how sensitive is the overall reconstruction performance to inaccuracies in the SDF, and how does SDF quality impact the fidelity of the extracted polygon vertices?

---
[A] Deep Unsupervised Learning using Nonequilibrium Thermodynamics

[B] Generative Modeling by Estimating Gradients of the Data Distribution

[C] Denoising Diffusion Probabilistic Models

---

> ### Author Rebuttal · Authors · 2025-07-29
>
> We thank the reviewer for their overall positive evaluation and helpful feedback for our work. We also appreciate their recognition of our use of SDF as an intermediate representation. Below, we address the reviewer’s questions and concerns about the SDF to polygon generation error propagation, characterization quantitative performance, triangulation vs visibility graph differentiation, references addition and practical application-
>   - **SDF to polygon generation error propagation**: The reviewer highlighted the absence of SDF to Polygon Extraction Error Propagation ablation study. To address this, we analyze the impact of perturbations in the SDF on the downstream polygon generation quality. Specifically, we report the change in performance metrics when varying levels of noise are added to the SDF input. We observe that as the SDF becomes increasingly distorted, the accuracy of vertex extraction consistently degrades, since the zero-level set becomes harder to localize and fails to form a well-defined polygon consistent with the visibility graph.
>
>     | SDF Noise Standard Deviation | F-1 Score |
>     |------------------------------|-----------|
>     | 0.0                          | **0.912** |
>     | 0.01                         | **0.912** |
>     | 0.05                         | 0.909     |
>     | 0.1                          | 0.905     |
>     | 0.5                          | 0.859     |
>   - **Characterization coverage and diversity quantitative performance**: The reviewer rightly expressed concern that there is no performance measure defined for the characterization problem. We thank the reviewer for this valuable suggestion. Below we share some of our insights regarding why quantifying the diversity and coverage of the characterization problem is a difficult task, and some initial steps we took toward tackling it
>     - First we note that diversity and the completeness of the solution are not identical. There might be cases where the complete solution may not be diverse (it could even contain a single polygon).
>     - Second, choosing a diversity metric is not straightforward. For example, let’s say the input graph is a clique $K(n)$. Then the solution set is the set of n-vertex convex polygons. if we adopt distribution of edge-lengths or angles as a diversity measure, it would show high diversity but arguably the solution is not combinatorially diverse.
>     - Still, we agree with the reviewer that this is an important matter. Towards addressing it, we started investigating the use of Chamfer distance for diversity and also a new approach where we pose the coverage problem as a type of metric space exploration problem:
>     - **Diversity**:
>         - Given a visibility graph $G$, sample $N$ polygons for that visibility graph and filter out a set $X$ of valid polygons using the recognition approach described in Section 6.4
>         - For all pairs of polygons within $X$ determine the chamfer distance between the polygons points set
>
>        We choose the chamfer distance as a diversity measure as it captures geometric differences through boundary misalignment and is also invariant to vertex ordering.  We achieve **$\mathbf{0.5568 \pm 0.1416}$** chamfer distance across the test dataset below for $N$ = 50. Given that all polygons lie within a 2×2 bounding box, a mean Chamfer distance of 0.56 corresponds to 20% of the domain's spatial extent, indicating significant geometric diversity.
>     - **Coverage**: The calculation of coverage requires enumeration of the solution space. However, the space of polygon to visibility graphs is not enumerable, making direct coverage evaluation intractable. Additionally, it is also not a well-defined metric space. Hence, it is difficult to define a notion of neighborhood that can be used to compute a coverage metric. Hence, we propose use of an exploration algorithm-
>         - Given a visibility graph G, we initialize a root polygon $P_{0}$ by sampling a latent noise vector with a base standard deviation $\sigma$ and generating a polygon using VisDiff.
>         - We then perform Breadth-First Exploration up to a fixed depth $d$ and branching factor b where each child is generated by adding scheduled noise to its parent in the latent space. Lastly, the nodes are expanded according to the following criteria:
>             - The generated polygon must be valid, and its recognition F-1 score with respect to the input visibility graph $G$ must exceed a threshold $T$ (Please refer to Recognition Approach in Section 6.4).
>             - The polygon must be distinct from previously discovered nodes, based on a distance threshold $T_{d}$​.
>         - We define the $\text{Coverage Metric} = \frac{\text{Number of Expanded Nodes}}{\text{Max Possible Nodes}}$ which quantifies the amount of search space covered.
>
>         This metric reflects the proportion of the latent neighborhood explored that yields valid, diverse polygons consistent with the input visibility graph. We now report the average coverage metrics across the test dataset for different hyperparameters. It can be observed that, given different hyper-parameters for $T$, we are expanding ~51% of the nodes which is ~32 nodes out of the maximum 63 possible. This is high, considering the training dataset contains only 20 augmentations (Different polygon, same visibility graph) per visibility graph. This suggests that our latent exploration strategy is able to discover a broader set of valid solutions than those during training.
>
>         | F-1 Threshold $T$ | Depth $d$ | Branching Factor $b$ | Distance Threshold $T_d$ | Coverage Metric |
>         |-------------------|-----------|------------------------|---------------------------|------------------|
>         | 0.85              | 5         | 2                      | 0.1                       | 0.475            |
>         | 0.80              | 5         | 2                      | 0.1                       | 0.488            |
>         | 0.75              | 5         | 2                      | 0.1                       | 0.495            |
>         | 0.70              | 5         | 2                      | 0.1                       | 0.515            |
>
>        We thank the reviewer again for their comments initiating this interesting line of inquiry. We plan to build on these results in our future work.
>   - **Triangulation graph vs visibility**: The reviewer expressed confusion about the importance of the triangulation section. To address the reviewer’s comment, we would like to mention that the visibility graph encodes a complete set of geometric relationships while the triangulation graph is a subset which provides sparser local structure, making the reconstruction problem more difficult. The triangulation subsection thus highlights VisDiff’s ability to generate polygonal shapes even when only local combinatorial structure is available, showing the model's generality to different combinatorial structures.
>   - **Demonstration of Practical Application**: We thank the reviewer for highlighting the importance of demonstrating practical applicability. In this work, we aimed to look at it from the perspective of learning combinatorial structures lacking well-behaved local neighborhoods or distance functions, and showing that existing state-of-the-art approaches fail to effectively capture these structures. In the future, we plan to leverage this finding to address applications such as privacy-aware floorplan generation. Current floorplan generation approaches [*1*, *2*] use polygons as their fundamental representation and typically rely only on the floorplan boundary and room connectivity for automatic generation. Room to room visibility (commonly referred as visibility graph analysis [*3*]) is not considered in these methods, despite being a critical factor in real-world floorplan design for incorporating occupant privacy requirements. Hence, we plan to leverage our findings to account for room-to-room visibility in floorplan generation approaches.
>
>     **References**
>
>     [*1*] Hong, Shibo, et al. *Cons2Plan: Vector Floorplan Generation from Various Conditions via a Learning Framework based on Conditional Diffusion Models*. Proceedings of the 32nd ACM International Conference on Multimedia, 2024.
>
>     [*2*] Shabani, Mohammad Amin, Sepidehsadat Hosseini, and Yasutaka Furukawa. *HouseDiffusion: Vector Floorplan Generation via a Diffusion Model with Discrete and Continuous Denoising*. Proceedings of the IEEE/CVF Conference on Computer Vision and Pattern Recognition, 2023.
>
>     [*3*] Park, Keundeok, Semiha Ergan, and Chen Feng. *Quality Assessment of Residential Layout Designs Generated by Relational Generative Adversarial Networks (GANs)*. Automation in Construction 158 (2024): 105243.
> - **References Addition**: We appreciate the reviewer’s suggestion to include fundamental references on diffusion. We apologize for this omission and will ensure that the relevant citations are added in the final camera-ready version of the paper.
>
> We would like to finally thank the reviewer for their time and efforts in reviewing our work and giving constructive feedback. The reviewer’s feedback with respect to quantifying the performance of the characterization problem has opened exciting directions of research. We intend to explore these aspects in future work and believe they can significantly strengthen the scope and impact of our work. If there are any additional questions or clarifications, we would be happy to address them.

---

> > ### Comment · Reviewer_Z2o4 · 2025-08-04
> >
> > Thank you for the detailed rebuttal.
> >
> > The "SDF to polygon generation error propagation" results align with the natural understanding of the two-stage pipeline and would be a valuable complement to the experiments.
> >
> > The new studies on the coverage and diversity are very interesting and informative. Please incorporate them concisely into the paper as a study of the characterization problem.
> >
> > My concerns from the initial review have been resolved, and I will increase my score to 5.

---

> > > ### Author Response · Authors · 2025-08-07
> > >
> > > Thank you for taking the time to engage with the rebuttal.
> > > We will incorporate the SDF-to-Polygon error propagation results and the coverage/diversity studies into the final version as part of the characterization analysis.
> > > We appreciate your valuable suggestions and support throughout the review process.

---

### Official Review · Reviewer_16i1 · 2025-07-03

**Clarity:** 3
**Significance:** 2
**Originality:** 3
**Rating:** 5
**Confidence:** 3

**Summary:**

- The paper presents an approach for 2D polygon generation given a corresponding 2D visibility graph, occupancy graph, or adjacency matrix.

- Three types of experiments are conducted: (1) reconstruction (generating a single polygon from an adjacency matrix), (2) generation of all plausible polygons, and (3) recognition (determining whether a valid polygon exists for a given visibility graph).

- A dense set of visual results is provided in the supplementary material.

**Questions:**

Majority of the weakness section points are minor questions to the authors. Would appreciate if they can answer them during the rebuttal phase.

**Ethical Concerns:**

["NO or VERY MINOR ethics concerns only"]

**Final Justification:**

The authors addressed my queries and the paper initially also had good contributions. I will keep my stance of the paper getting accepted.

**Limitations:**

yes

**Quality:**

3

**Strengths And Weaknesses:**

Strengths
- The paper is well written and easy to follow.
- +1 for the OOD test examples evaluation.
- Lot of visual comparison and good quantitative analysis

Weaknesses
- After SDF generation, one possible baseline for polygon creation could be a non-learned method such as the marching cubes algorithm. It would be interesting if the authors could comment on this approach and, if feasible, provide a comparison.

- An alternative worth considering is learned marching cubes — see the Deep Marching Cubes paper.

- It would also be valuable to hear the authors' thoughts on a joint pipeline for SDF denoising, SDF encoding, and vertex prediction. If such an approach was attempted and found ineffective, including that insight would be helpful. Recent works like REPA-E have demonstrated the benefits of jointly training vision encoders (e.g., VAEs) with latent diffusion models, resulting in significant speedups.

- It would have been useful if the paper included results for both 2D and 3D polygon generation.

- It would also be interesting to know whether the authors experimented with unconditional denoising and generation of SDF maps, allowing polygon generation at test time via SDF denoising for arbitrary visibility graphs.

- For figures like Figure 3, it would have been helpful to overlay the generated polygon on the visibility maps for clearer visual interpretation.

---

> ### Author Rebuttal · Authors · 2025-07-29
>
> We thank the reviewer for their overall positive evaluation and helpful feedback for our work. We also appreciate their recognition of our OOD evaluation. Below, we address the reviewer’s questions and concerns about the comparison with other polygon extraction approaches, joint training of an SDF denoising-encoding pipeline with vertex prediction, unconditional SDF-diffusion with conditional vertex generation results, 3D results and Figure 3 changes-
> - **Polygon Extraction**: We thank the reviewer for their suggestion regarding Marching Cubes and Deep Marching Cubes as alternative polygon extraction methods. We provide a comparison between our polygon extraction approach and the standard Marching Cubes method below. Standard Marching Cubes did not work well as it operates purely on local threshold crossings and does not consider the visibility graph. This results in rounded boundaries, missing sharp corners causing the reconstructed polygon to violate the visibility constraints. We appreciate the mention of Deep Marching Cubes. While it was not explored in the current version, we will plan to investigate it in our future work.
> | Experiment                       | F-1 Score |
> |----------------------------------|-----------|
> | With Vertex Prediction Architecture | **0.912** |
> | Marching Cubes Algorithm         | 0.800     |
> - **Joint training of an SDF denoising-encoding pipeline with vertex prediction**: We appreciate the interest of the reviewer for the insight with respect to joint training of the SDF denoising-encoding pipeline with vertex prediction. We initially experimented with an end-to-end (SDF denoising-encoding and polygon extraction) training. Our experiment showed that initializing the polygon reconstruction with the contour of the SDF generated by the trained SDF diffusion model leads to a notable improvement in performance metrics. We provide a comparison between joint training and our two-stage training approach that includes polygon contour initialization-
> | Experiment                                                                 | F-1 Score |
> |----------------------------------------------------------------------------|-----------|
> | Joint training SDF denoising-encoding pipeline with vertex prediction     | 0.850     |
> | Two-Stage With Vertex Initialization                                      | **0.912** |
> - **Unconditional denoising and conditional vertex generation**: We appreciate the reviewer’s interest in the results of our experiment involving unconditional SDF generation followed by conditional polygon reconstruction. We provide a quantitative comparison between conditional SDF and polygon generation, and unconditional SDF with conditional polygon generation. A noticeable drop in performance is observed when compared to conditional SDF and polygon generation, indicating that reconstruction quality strongly depends on the quality of the prior conditional SDF generation.
> | Experiment                                         | F-1 Score |
> |----------------------------------------------------|-----------|
> | Conditional SDF + Polygon Generation               | **0.912** |
> | Unconditional SDF + Conditional Polygon Generation | 0.865     |
> We thank the reviewer again for their comments about this line of ablation study which has given us further insights into the approach.
> - **3D Results**: The reviewer was interested in seeing the results of 3D polygons. We agree with this valuable direction of applying the approach to 3D. We focused on the 2D case in our current work to maintain a clear scope. Visibility gets significantly more complex and sometimes counter-intuitive in 3D. For example, in 3D, seeing all vertices does not guarantee seeing the interior! Extending to 3D is an exciting avenue we plan to explore in future work. Thank you for highlighting this potential.
> - **Figure 3 Change**: We thank the reviewer for suggesting a better way to show Figure 3. We will incorporate this change in the final camera-ready version of the paper.
>
> We would like to again thank the reviewer again for their time and efforts in reviewing our work and giving constructive feedback.  The reviewer’s suggestion with respect to adding additional ablation study of unconditional SDF generation with conditional polygon generation has been really helpful as it has given us additional insights into the approach. Furthermore, we acknowledge the reviewer’s suggestion to better represent Figure 3 which we will include in the camera-ready version. Please let us know if there are any further questions or clarifications, we would be happy to address them.

---

> > ### Author Response · Authors · 2025-08-07
> > **Follow Up**
> >
> > We sincerely thank the reviewer once again for their insightful feedback and recognition of key aspects of our work. We wanted to follow up in case you had any additional thoughts or needed clarification on any part of our response. Your input on areas like ablation studies has been valuable in strengthening our work. We're happy to engage further if there are any further questions or clarifications.

---

> > ### Comment · Reviewer_16i1 · 2025-08-07
> > **Response to Author's Rebuttal**
> >
> > Firstly, I appreciate the authors’ efforts in thoroughly addressing my questions. I have already voted to accept the paper, and in light of their responses, I will maintain my stance.
> >
> > While I believe that improved training strategies could make joint training perform better, I agree that a thorough investigation of this is beyond the scope of the rebuttal. The authors' attempt to make joint training work already represents a meaningful step in that direction.
> >
> > Best Regards!

---

### Official Review · Reviewer_nwcz · 2025-07-03

**Clarity:** 4
**Significance:** 3
**Originality:** 3
**Rating:** 4
**Confidence:** 4

**Summary:**

The paper considers the task of 2D polygon generation conditioned on the input desired visibility graph of the polygon. The authors propose a two-stage method as a potential solution for the task. The method treats visibility graphs as grayscale 2D images and uses image-conditioned 2D diffusion to first generate 2D grids of SDF values. In the second step, initial polygons are extracted from the sampled SDF grids with contour extraction methods, and the initial vertices, combined with the global and local features, are further refined into the final vertex positions.

Visibility graphs do not fully define the corresponding polygons, since one graph can correspond to multiple polygons. To perform the experiments, the authors propose a synthetic dataset that was constructed by first sampling the polygons, filtering the set to ensure uniform link diameter distribution, and augmenting the polygons corresponding to each visibility graph to obtain multiple valid polygons for a single graph.

The approach is trained and evaluated on the proposed dataset and compared to several graph and mesh generation baselines adapted for the task. The authors use classification-based metrics for evaluation and comparison (after the polygon is sampled, its visibility graph is constructed and compared to the input visibility graph).

The proposed method outperforms the baselines. It is additionally demonstrated that it can sample multiple different polygons for a single input visibility graph. The authors also explore the possibility of using their model as a classification method for the detection of invalid visibility graphs, and for various interpolation applications.

**Questions:**

The idea of using SDF as an intermediate representation for mesh extraction has been explored for multiple years by the differentiable meshing community (e.g. MeshSDF, Neural Dual Contouring, and many others). It is possible to go from SDF to polygons in a differentiable, or at least partially differentiable, manner. Have the authors considered that line of work?

What is the reason behind the use of only 25-vertex polygons? It may be easier to process the data if all the polygons are of the same size, but there are numerous mesh-based deep learning methods capable of working with variable-size meshes. Adapting the architecture and computational graphs for it should not be a big issue nowadays. Working with more diverse data would look much more convincing.

**Ethical Concerns:**

["NO or VERY MINOR ethics concerns only"]

**Final Justification:**

After the rebuttal, I decided to keep my original rating and overall support the acceptance of the paper. I still believe that the task of conditional graph generation given visibility information is quite novel, and the proposed method is a good early attempt at solving the task. On the other hand, some of the original weak points (no guarantees of corresponding visibility in generated results, limited implementation working with fixed-size graphs) remain even after clarifications and consideration of the provided additional results.

**Limitations:**

The paper does not discuss the limitations. I would at least mention no guarantees of correspondence of sampled polygons to the input visibility graphs and the limited nature of evaluation (limited diversity of data, only synthetic data).

**Paper Formatting Concerns:**

No concerns.

**Quality:**

3

**Strengths And Weaknesses:**

Pros:
1. The paper considers a relatively underexplored problem, introduces it in an efficient manner, and is well-written.
2. The proposed method seems to be a valid groundbreaking approach for the task.
3. The method is supported by a good number of experiments.

Cons:
1. My main critique of the approach is its completely data-driven nature. The authors propose to fully rely on the model's intrinsic ability to capture the diversity of the possible polygon configurations corresponding to a single visibility graph. There are no consistency mechanisms enforcing the model to output polygons corresponding to the input graph, except for the data-driven training loss. In principle, modern advances in deep learning demonstrate that given a sufficiently large model and a dataset as diverse as possible (the proposed dataset is definitely not very diverse, there are only polygons with 25 vertices in it), it may be enough, but it would be interesting to see if any explicit consistency mechanism can boost the performance.
2. The paper could benefit from additional results on either a more diverse synthetic dataset (polygons with variable vertex cardinality), or some real data (semantic segmentation datasets with polygon-based masks exist; maybe it is possible to use this data for some additional experiments).

---

> ### Author Rebuttal · Authors · 2025-07-29
>
> We thank the reviewer for their overall positive evaluation and helpful feedback of our work. We also appreciate their recognition of our approach being ground-breaking. Below, we address the reviewer’s questions and concerns about the dataset, architecture end-to-end differentiability, consistency mechanism and limitation section-
> - **Dataset**: The reviewer asks the reasoning behind the choice of 25 as the number of vertices (Q2) and also suggests using real data such as polygons representing  segmentation masks as potentially beneficial for the paper in terms of diversity (W2).
>     - **Diversity of dataset (W2)**: As the reviewer suggests,  using real datasets will result in higher diversity in terms of vertex cardinality. However, we show below that our synthetic dataset has much **higher structural diversity** compared to real datasets such as MNIST (converted to polygonal contours) and COCO 2017 (object masks) with respect to the combinatorial properties of the visibility graph such as link diameter. (link diameter captures the level of concavity of the polygon.)
> | Dataset    | Link Diameter (Avg / Std) | Min  | Max   |
> |------------|----------------------------|------|-------|
> | MNIST      | 1.83 / 0.50                | 1.0  | 4.1   |
> | COCO 2017  | 1.32 / 0.25                | 1.0  | 3.823 |
> | VisDiff    | **4.4 / 2.2**              | 1.0  | **9.0** |
>   - **Choice of 25 for the polygon size (number of vertices)**: We thank the reviewer for this comment. First, we note that varying the number of vertices is only indirectly related to structural diversity. Instead, the maximum number of vertices is the key factor as one can simply pad the boundary to reach a fixed maximum number, without changing the overall shape. Therefore, the main question is why we picked 25 as the number of vertices. The short answer is resource availability, in particular memory. Increasing the number of points requires a finer grid to capture small visibility changes. A finer grid results in a significant increase in computational demands. In particular, we trained our model on 1 v100 GPU with 30 GB RAM. The SDF diffusion model with a grid of 0.05 resolution (grid size for 25 vertex locations) uses 10 GB of memory during training. Reducing the grid resolution to 0.01(grid size for 50 vertex locations), the memory consumption increases to 40 GB. As mentioned in the conclusion, we plan to remove the grid structure entirely to overcome this limitation and further extend the dataset to a varying number of vertex locations.
> - **Architecture end-to-end differentiability and consistency mechanism**: We appreciate the interest of the reviewer to explore end-to-end differentiable architecture for polygon extraction and additional consistency mechanism for performance boost.
>   - **End-to-End Differentiable**: We initially experimented with an end-to-end (SDF-generation and polygon extraction) fully differentiable pipeline. Our experiment showed that initializing the polygon reconstruction with the contour of the SDF generated by the trained SDF diffusion model leads to a notable improvement in performance metrics. We provide below a comparison between end-end differentiable training of SDF denoising-encoding pipeline with vertex prediction and our two-stage training approach that includes polygon contour initialization.
> | Experiment                                                                 | F-1 Score |
> |----------------------------------------------------------------------------|-----------|
> | End-to-End Diffrentiable training SDF denoising-encoding pipeline with vertex prediction     | 0.850     |
> | Two-Stage With Vertex Initialization                                      | **0.912** |
>   - **Consistency Mechanism**: We appreciate the reviewer’s concern for further consistency approaches to boost the performance of the polygon generation based on visibility. In fact, we started our investigation with a similar optimization approach using custom visibility and crossing losses computed from the intermediate predicted SDF representation. We observed no notable boost in performance and therefore did not include the approach in the paper.
> - **Limitation Section**: The reviewer mentions that no limitations were discussed in the paper. To clarify, we do discuss the limitations of our approach in the conclusion section, noting that it is bottle-necked by computation time and memory. We also outline future directions aimed at mitigating these challenges. However, we agree with the limitation raised by the reviewer that we have no guarantees of 100% correspondence of the sampled polygons to the input visibility graphs. We thank the reviewer for prompting us to mention it explicitly. Hence, we will make sure to elaborate on this point in greater detail in the final camera-ready version of the paper.
>
> We would like to once again thank the reviewer for their time and efforts in reviewing our work and giving helpful suggestions. We value the reviewer’s interest in exploring consistency mechanisms to further boost the performance and to explicitly discuss additional limitations to further strengthen the paper. We would be happy to make our detailed optimization approach available in the arxiv version after the review period ends and would discuss these additional limitations in the camera-ready version of the paper. Please let us know if there are any further questions or clarifications, we would be happy to address them.

---

> > ### Author Response · Authors · 2025-08-07
> > **Follow Up**
> >
> > Thank you again for your thoughtful and constructive feedback. We wanted to follow up on our rebuttal in case you had any further questions or suggestions. Your comments, especially regarding consistency mechanisms and architectural choices, were very helpful, and we would be happy to clarify or expand on any part of our response.

---

> > ### Comment · Reviewer_nwcz · 2025-08-07
> >
> > Thank you for the provided answers and clarifications.
> >
> > I understand and agree with your argument about the custom dataset having more structural diversity. It is clear that since it is synthetic, you can produce a richer set of possible visibility graph configurations. At the same time, I would like to underline that the performance on real graphs is as important as the performance on the synthetic data, even if it is less diverse in a formal sense, because it demonstrates the potential of the method when applied to real data (which is usually the ultimate objective of any applied method).
> >
> > Thank you for the clarification about the memory limitations of the method. It might make sense to mention these constraints in the limitations section. It would be interesting to see how future extensions will circumvent the memory efficiency problems, as well as the reliance on a fixed number of vertices.
> >
> > As for the differentiable pipeline experiment, it is confusing to me. I assume that this baseline was in some sense "weaker" than the proposed method (different networks used, different parameter capacity). Maybe I missed something, but it is hard for me to believe that a method that can be overfitted to match any sample almost perfectly (based on differentiability) can be easily outperformed by a method that is guessing a solution from an input image. However, I do not think that the authors should be punished because of the absence of such an experiment in their submission. If the authors plan to include such an experiment in the final version of their paper, I would suggest providing a detailed description of the baseline based on end-to-end differentiable training.

---

> ### Author Response · Authors · 2025-08-07
>
> Thank you for engaging with our rebuttal and for your valuable comments.
>
> We acknowledge the reviewer’s points regarding the importance of evaluating on real data and the confusion surrounding the joint differentiable training pipeline. Accordingly, we will include both the real-data results and a detailed description of the joint differentiable baseline architecture and training setup in the arXiv version of the paper.
>
> Lastly, we will also explicitly mention the memory efficiency constraints in the limitations section of the camera-ready version.
>
> We appreciate your valuable suggestions and support throughout the review process.

---

### Comment · Area_Chair_1GCc · 2025-08-05

Dear reviewers,

Thanks for those of you who already engaged with the rebuttal provided by the authors. I ask reviewers `nwcz` and `16i1` who have not yet done so to **respond to the rebuttal** and ask any followup questions they may have. Please note that you need to **acknowledge** participation in the rebuttal process and submit a final recommendation, which will be hidden to the authors unless you choose to divulge it. Given the additional time for rebuttal discussions, make sure to **engage** and provide feedback to the authors so that they may improve their manuscript moving forward.

Thanks!

  — Your AC

---

### Note · Authors · 2025-08-15

We would like to thank the area chairs and the reviewers for their time and expertise. We sincerely appreciate the reviewer’s insights, constructive suggestions, and valuable comments. Below, we provide an overall summary of the discussions and the additional experiments we conducted to address the reviewer’s comments:

$\textbf{Summary of Discussion}$

* $\textbf{Diversity of Dataset}$: We compared our dataset to real datasets such as MNIST and COCO segmentation masks in terms of the link diameter of the visibility graph. Our synthetic dataset demonstrates higher structural diversity than existing datasets (requested by reviewers [nwcz](https://openreview.net/forum?id=8M9T7Nl454&noteId=kvVoFMEI2K) and [UnHa](https://openreview.net/forum?id=8M9T7Nl454&noteId=7sajujwdtT)).
* $\textbf{SDF Error Propagation}$: To examine the effect of SDF errors on vertex generation, we provided the following additional results:
  * SDF Error vs Polygon Generation performance (requested by reviewer [Z2o4](https://openreview.net/forum?id=8M9T7Nl454&noteId=mA6xhTvT9q))
  * Unconditional SDF Generation with Visibility conditioned Vertex Generation (requested by reviewer [16i1](https://openreview.net/forum?id=8M9T7Nl454&noteId=FQfmnKFuFt))
* $\textbf{Characterization Coverage and Diversity Quantitative Performance}$:  We proposed an approach to determine diversity and coverage for the characterization problem, and provided quantitative results in the rebuttal showing that our method achieves strong performance on both metrics. (requested by reviewer [Z2o4](https://openreview.net/forum?id=8M9T7Nl454&noteId=mA6xhTvT9q))

Thank you again for your time and feedback.

---

### Decision · Program_Chairs · 2025-09-17

**Decision:**

Accept (poster)

**Comment:**

# Summary

This paper provides a novel diffusion-based approach to reconstruct polygons from their visibility graphs. An understudied problem in computational geometry, the paper presents a method for predicting a signed distance function (SDF) condition on visibility graph. In addition to this methodological contribution, the authors also provide a novel, curated dataset, demonstrating the utility of their method in comparison to others.

# Strengths

- A novel, interesting, and somewhat understudied problem with numerous potential downstream applications
- Predicting the SDF as an intermediate representation is a smart design choice and highly innovative
- The paper's methodological contributions are complemented by strong empirical evidence, and the curated dataset is highly relevant as its own contribution

# Weaknesses

- Some minor concerns about the generality of the method
- Some missing references and related work that could be better contextualized
- Limited discussion of practical utility in the first version of the text; the rebuttal served to provide clarifications here

# Decision and rationale

Despite some (arguably minor!) limitations in scope, this work is highly innovative and could potentially open up new avenues of research in geometric deep learning or combinatorial learning. With the main method being technically novel, justified, and well-validated, all reviewers unanimously suggested acceptance. Given the strong rebuttal by the authors, I am happy to second this endorsement and suggest _accepting_ this work.

# Summary of the discussion

- Reviewer `nwcz`: Raised some concerns about the data-driven nature of the method and suggested more datasets; reviewer did not provide a final justification, but the AC found the rebuttal to be more than sufficient.

- Reviewer `16i1`: Maintained a positive stance; asked about additional polygon extraction methods; authors provided clarifications in th rebuttal, which the reviewer acknowledged.

- Reviewer `Z2o4`: Raised some concerns about references and lack of certain analyses; the authors were able to address all these concerns in their rebuttal, prompting the reviewer to endorse the paper for publication.

- Reviewer `UnHa`: Raised concerns about experimental evaluations; authors addressed these concerns, and the reviewer decided to endorse the paper for publication.

In summary, this submission amply demonstrates the utility of a strong rebuttal—all reviewers essentially agreed that this paper is ready for publication.